# Locality defeats the curse of dimensionality in convolutional teacher-student scenarios

**Alessandro Favero** *
Institute of Physics
École Polytechnique Fédérale de Lausanne
`alessandro.favero@epfl.ch`

**Francesco Cagnetta** *
Institute of Physics
École Polytechnique Fédérale de Lausanne
`francesco.cagnetta@epfl.ch`

**Matthieu Wyart**
Institute of Physics
École Polytechnique Fédérale de Lausanne
`matthieu.wyart@epfl.ch`

## Abstract

Convolutional neural networks perform a local and translationally-invariant treatment of the data: quantifying which of these two aspects is central to their success remains a challenge. We study this problem within a teacher-student framework for kernel regression, using 'convolutional' kernels inspired by the neural tangent kernel of simple convolutional architectures of given filter size. Using heuristic methods from physics, we find in the ridgeless case that locality is key in determining the learning curve exponent $\beta$ (that relates the test error $\epsilon_t \sim P^{-\beta}$ to the size of the training set $P$), whereas translational invariance is not. In particular, if the filter size of the teacher $t$ is smaller than that of the student $s$, $\beta$ is a function of $s$ only and does not depend on the input dimension. We confirm our predictions on $\beta$ empirically. We conclude by proving, under a natural universality assumption, that performing kernel regression with a ridge that decreases with the size of the training set leads to similar learning curve exponents to those we obtain in the ridgeless case.

## 1 Introduction

Deep Convolutional Neural Networks (CNNs) are widely recognised as the engine of the latest successes of deep learning methods, yet such a success is surprising. Indeed, any supervised learning model suffers *in principle* from the curse of dimensionality: under minimal assumptions on the function to be learnt, achieving a fixed target generalisation error $\epsilon$ requires a number of training samples $P$ which grows exponentially with the dimensionality $d$ of input data [1], i.e. $\epsilon(P) \sim P^{-1/d}$. Nonetheless, empirical evidence shows that the curse of dimensionality is beaten *in practice* [2, 3, 4], with

$$\epsilon(P) \sim P^{-\beta}, \quad \beta \gg 1/d. \tag{1}$$

CNNs, in particular, achieve excellent performances on high-dimensional tasks such as image classification on ImageNet with state-of-the-art architectures, for which $\beta \approx [0.3, 0.5]$ [2]. Natural data must then possess additional structures that make them learnable. A classical idea [5] ascribes the success of recognition systems to the compositionality of data, i.e. the fact that objects are made of features, themselves made of sub-features [6, 7, 8]. In this view, the locality of CNNs plays a key role for their performance, as supported by empirical observations [9]. Yet, there is no clear

---

*Equal contribution.

35th Conference on Neural Information Processing Systems (NeurIPS 2021).

analytical understanding of the relationship between the compositionality of the data and learning curves.

In order to study this relationship quantitively, we introduce a teacher-student framework for kernel regression, where the function to be learnt takes one of the following two forms:

$$f^{LC}(\boldsymbol{x}) = \sum_{i \in \mathcal{P}} g_i(\boldsymbol{x}_i), \quad f^{CN}(\boldsymbol{x}) = \sum_{i \in \mathcal{P}} g(\boldsymbol{x}_i). \tag{2}$$

Here, $\boldsymbol{x}$ is a $d$-dimensional input and $\boldsymbol{x}_i$ denotes the $i$-th $t$-dimensional patch of $\boldsymbol{x}$, $\boldsymbol{x}_i = (x_i, \ldots, x_{i+t-1})$. $i$ ranges in a subset $\mathcal{P}$ of $\{1, \ldots, d\}$. The $g_i$'s and $g$ are random functions of $t$ variables whose smoothness is controlled by some exponent $\alpha_t$. Such functions model the local nature of certain datasets and can be generated, for example, by randomly-initialised one-hidden-layer neural networks: $f^{LC}$ corresponds to a *locally connected* network (LCN) [10, 11], in which the input is split into lower-dimensional patches before being processed, whereas a network enforcing invariance with respect to shifts of the input patches via weight sharing can be described by $f^{CN}$. In such cases $t$ would be the filter size of the network. Our goal is to compute the asymptotic decay of the error of a student kernel performing regression on such data, and to relate the corresponding exponent $\beta$ to the locality of the target function. The student kernel corresponds to a prior on the true function of the form described by Eq. (2), except that the filter size $s$ and its prior $\alpha_s$ on the smoothness of the $g$ functions can differ from those of the target function. Such students include overparametrised one-hidden-layer neural networks operating in the *lazy training regime* [12, 13, 14, 15, 16].

## 1.1 Our contributions

We consider a teacher-student framework for kernel regression, where the target function has one of the forms in Eq. (2), where the $g_i$'s and $g$ are Gaussian random fields of given covariance. Target functions are characterised by the dimensionality $t$ of the $g$ functions—the *filter size*—and a smoothness exponent $\alpha_t$, such that $\alpha_t > 2n$ implies that typical target functions are at least $n$ times differentiable. Kernel regression is performed by *local* or *convolutional* student kernels, having filter size $s$ and a prior on the target smoothness characterised by another exponent $\alpha_s > 0$. Our main contributions follow:

- ○ We use recent results based on the replica method of statistical physics on the generalisation error of kernel methods [17, 18, 19] to estimate the exponent $\beta$. We find that $\beta = \alpha_t/s$ if $t \leq s$ and $\alpha_t \leq 2(\alpha_s + s)$. This approach is non-rigorous, but it can be proven if data are sampled on a lattice [4] and corresponds to a provable lower bound on the error when teacher and student are equal [20].

- ○ In particular, we find the same exponent for students with a prior on the shift invariance of the target function and students without this prior, implying that the curse of dimensionality is beaten due to locality and not shift invariance.

- ○ We confirm systematically our predictions by performing kernel ridgeless regression numerically for various $t$, $s$ and embedding dimension $d$.

- ○ We use the recent framework of [21] and a natural Gaussian universality assumption to prove a rigorous estimate of $\beta$ in the case where the ridge decreases with the size of the training set. The estimate of $\beta$ depends again on $s$ and not on $d$, demonstrating that the curse of dimensionality can indeed be beaten by using local filters on such compositional data.

## 1.2 Related work

Several recent works study the role of the compositional structure of data [6, 22, 23]. When such structure is hierarchical, deep convolutional networks can be much more expressive than shallow ones [6, 24, 7]. Concerning training, [25] shows that both convolutional and locally-connected networks can achieve a target generalisation error in polynomial time, whereas fully-connected networks cannot, for a class of functions which depend only on $s$ consecutive bits of the $d$-dimensional input, with $s = \mathcal{O}(\log d)$. In [8] the effects of the architecture's locality are studied from a kernel perspective, using a class of deep convolutional kernels introduced in [26, 27] and characterising their Reproducing Kernel Hilbert Space (RKHS). In general, belonging to the RKHS ensures favourable bounds on performance and, for isotropic kernels, is a constraint on the function smoothness that becomes

stringent in large $d$. For local functions, the corresponding constraint on smoothness is governed by the filter size $s$ and not $d$ [8]. Lastly, a recent work shows that weight sharing, in the absence of locality, leads to a mild improvement of the generalisation error of shift-invariant kernels [28].

By contrast, our work focuses on computing non-trivial training curve exponents in a setup where the locality and shift-invariance priors of the kernel can differ from those of the class of functions being learnt. In our setup, the latter are in general not in the RKHS of the kernel[2]. Technically, our result that the size of the student filter $s$ controls the learning curve (and not that of the teacher $t$) relates to the fact that kernels are not able to detect data anisotropy (the fact that the function depends only on a subset of the coordinates) in worst-case settings [30] nor in the typical case for Gaussian fields [31].

## 2  Setup

**Kernel ridge regression**   Kernel ridge regression is a method to learn a target function $f^* : \mathbb{R}^d \to \mathbb{R}$ from $P$ observations $\{(\boldsymbol{x}^\mu, f^*(\boldsymbol{x}^\mu))\}_{\mu=1}^P$, where the inputs $\boldsymbol{x}^\mu$ are i.i.d. random variables distributed according to a certain measure $p\left(d^d x\right)$ on $\mathbb{R}^d$. Let $K$ be a positive-definite kernel and $\mathcal{H}$ the corresponding Reproducing Kernel Hilbert Space (RKHS). The kernel ridge regression estimator $f$ of the target function $f^*$ is defined as

$$f = \underset{f \in \mathcal{H}}{\operatorname{argmin}} \left\{ \frac{1}{P} \sum_{\mu=1}^P (f(\boldsymbol{x}^\mu) - f^*(\boldsymbol{x}^\mu))^2 + \lambda \|f\|_{\mathcal{H}}^2 \right\}, \tag{3}$$

where $\| \cdot \|_{\mathcal{H}}$ denotes the RKHS norm and $\lambda$ is the ridge parameter. The limit $\lambda \to 0^+$ is known as the ridgeless case and corresponds to the solution with minimum RKHS norm that interpolates the $P$ observations. Eq. (3) is a convex optimisation problem, having the unique solution

$$f(\boldsymbol{x}) = \frac{1}{P} \sum_{\mu,\nu=1}^P K(\boldsymbol{x}, \boldsymbol{x}^\mu) \left( \left( \frac{1}{P}\mathbb{K}_P + \lambda \mathbb{I}_P \right)^{-1} \right)_{\mu,\nu} f^*(\boldsymbol{x}^\nu), \tag{4}$$

where $\mathbb{K}_P$ is the *Gram matrix* defined as $(\mathbb{K}_P)_{\mu\nu} = K(\boldsymbol{x}^\mu, \boldsymbol{x}^\nu)$, and $\mathbb{I}_P$ denotes the $P$-dimensional identity matrix. Our goal is to compute the generalisation error, which we define as the expectation of the mean squared error over the data distribution $p\left(d^d x\right)$, averaged over an ensemble of target functions $f^*$, i.e

$$\epsilon(P) = \mathbb{E}_{\boldsymbol{x}, f^*} \left[ (f(\boldsymbol{x}) - f^*(\boldsymbol{x}))^2 \right]. \tag{5}$$

The error $\epsilon$ depends on the number of samples $P$ through the predictor of Eq. (4) and we refer to the graph of $\epsilon(P)$ as *learning curve*.

**Statistical mechanics of generalisation in kernel regression**   The theoretical understanding of generalisation is still an open problem. A few recent works [17, 21, 18] relate the generalisation error $\epsilon$ to the decomposition of the target function in the eigenbasis of the kernel. A positive-definite kernel $K$ can indeed be written, by Mercer's theorem, in terms of its eigenvalues $\{\lambda_\rho\}$ and eigenfunctions $\{\phi_\rho\}$:

$$K(\boldsymbol{x}, \boldsymbol{y}) = \sum_{\rho=1}^\infty \lambda_\rho \phi_\rho(\boldsymbol{x}) \overline{\phi_\rho(\boldsymbol{y})}, \quad \int p\left(d^d y\right) K(\boldsymbol{x}, \boldsymbol{y}) \phi_\rho(\boldsymbol{y}) = \lambda_\rho \phi_\rho(\boldsymbol{x}). \tag{6}$$

In [17, 21, 18] it is shown that, when the target function can be written in terms of the kernel eigenbasis,

$$f^*(\boldsymbol{x}) = \sum_\rho c_\rho \phi_\rho(\boldsymbol{x}), \tag{7}$$

the error $\epsilon$ can also be cast as a sum of modal contributions, $\epsilon = \sum_\rho \epsilon_\rho$. The details of the general formulation are summarised in Appendix A. Here we present an intuitive limiting case, obtained in the ridgeless limit $\lambda \to 0^+$, when $\lambda_\rho \sim \rho^{-a}$ for large $\rho$, and $\mathbb{E}[|c_\rho|^2] \sim \rho^{-b}$ with $2a > b - 1$, that is

$$\epsilon(P) \sim \sum_{\rho > P} \mathbb{E}[|c_\rho|^2] \equiv \mathcal{B}(P), \tag{8}$$

---

[2]A Gaussian field of covariance $K$ is never in the RKHS of the kernel $K$, see e.g. [29].

with $\sim$ denoting asymptotic equivalence for large $P$. Eq. (8) indicates that, given $P$ examples, the generalisation error can be estimated as the tail sum of the power in the target function past the first $P$ modes of the kernel, which we denote as $\mathcal{B}(P)$. Although the general modal decomposition cannot be proven rigorously in the ridgeless limit [21, 19], additional results are available when the target functions are Gaussian random fields with covariance specified by a teacher kernel:

- Eq. (8) can be proven rigorously [4] if teacher and student are isotropic kernels and the input points $\boldsymbol{x}^\mu$ are sampled on the lattice $\mathbb{Z}^d$, i.e. all the elements of each input sequence are integer multiples of an arbitrary unit;
- If teacher and student coincide then $\mathbb{E}[|c_\rho|^2]$ equals the $\rho$-th eigenvalue $\lambda_\rho$ and (see e.g. [20]) $\epsilon(P) \geq \mathcal{B}(P)$, i.e. the estimate of Eq. (8) is a lower bound.

## 3    Kernels for local and convolutional teacher-student scenarios

In this section we introduce convolutional and local kernels that will be used as teachers, i.e. to generate different ensembles of target functions $f^*$ with controlled smoothness and degree of locality, and as student kernels. We motivate our choice by considering one-hidden-layer neural networks with simple local and convolutional architectures. Because of the relationship between our kernels and the Neural Tangent Kernel [12] of the aforementioned architectures, our framework encompasses regression with simple overparametrised networks trained in the lazy regime [16]. For the sake of clarity we limit the discussion to inputs which are sequences in $\mathbb{R}^d$, i.e. $\boldsymbol{x} = (x_1, \ldots, x_d)$. Extension to higher-order tensorial inputs such as images $\boldsymbol{X} \in \mathbb{R}^{d \times d}$ is straightforward. To avoid dealing with the boundaries of the sequence we identify $x_{i+d}$ with $x_i$ for all $i = 1, \ldots, d$.

**Definition 3.1** (one-hidden-layer CNN)**.** *A one-hidden-layer convolutional network with $H$ hidden neurons and average pooling is defined as follows,*

$$f^{CNN}(\boldsymbol{x}) = \frac{1}{\sqrt{H}} \sum_{h=1}^{H} a_h \frac{1}{|\mathcal{P}|} \sum_{i \in \mathcal{P}} \sigma(\boldsymbol{w}_h \cdot \boldsymbol{x}_i), \qquad (9)$$

*where $\boldsymbol{x} \in \mathbb{R}^d$ is the input, $H$ is the width, $\sigma$ a nonlinear activation function, $\mathcal{P} \subseteq \{1, \ldots, d\}$ is a set of patch indices and $|\mathcal{P}|$ its cardinality. For all $i \in \mathcal{P}$, $\boldsymbol{x}_i$ is an $s$-dimensional patch of $\boldsymbol{x}$. For all $h = 1, \ldots, H$, $\boldsymbol{w}_h \in \mathbb{R}^s$ is a filter with filter size $s$, $a_h \in \mathbb{R}$ is a scalar weight. The dot $\cdot$ denotes the standard Euclidean scalar product.*

In the network defined above, a $d$-dimensional input sequence $\boldsymbol{x}$ is first mapped to $s$-dimensional *patches* $\boldsymbol{x}_i$, which are ordered subsequences of the input. Comparing each patch to a filter $\boldsymbol{w}_h$ and applying the activation function $\sigma$ leads to a $|\mathcal{P}|$-dimensional hidden representation which is equivariant for shifts of the input. The summation over the patch index $i$ promotes this equivariance to full invariance, leading to a model which is both local and shift-invariant as $f^{CN}$ in Eq. (2). A model which is only local, as $f^{LC}$ in Eq. (2), can be obtained by lifting the constraint of weight-sharing, which forces, for each $h = 1, \ldots, H$, the same filter $\boldsymbol{w}_h$ to apply to all patches $\boldsymbol{x}_i$.

**Definition 3.2** (one-hidden-layer LCN)**.** *In the notation of Definition 3.1, a one-hidden-layer locally-connected network with $H$ hidden neurons is defined as follows,*

$$f^{LCN}(\boldsymbol{x}) = \frac{1}{\sqrt{H}} \sum_{h=1}^{H} \frac{1}{\sqrt{|\mathcal{P}|}} \sum_{i \in \mathcal{P}} a_{h,i} \sigma(\boldsymbol{w}_{h,i} \cdot \boldsymbol{x}_i), \qquad (10)$$

*For all $i \in \mathcal{P}$ and $h = 1, \ldots, H$: $\boldsymbol{x}_i$ is an $s$-dimensional patch of $\boldsymbol{x}$, $\boldsymbol{w}_{h,i} \in \mathbb{R}^s$ is a filter with filter size $s$, $a_{h,i} \in \mathbb{R}$ is a scalar weight.*

Notice that the definition above reduces to that of a fully-connected network when the filter size is set to the input dimension, $s = d$, and $\mathcal{P} = \{1\}$. With the target functions taking one of the two forms in Eq. (2), our framework contains the case where the observations are generated by neural networks such as (3.1) and (3.2). Let us now introduce the neural tangent kernels of such architectures.

**Definition 3.3** (Neural Tangent Kernel)**.** *Given a neural network function $f(\boldsymbol{x}; \boldsymbol{\theta})$, where $\boldsymbol{\theta} = (\theta_1, \ldots, \theta_N)$ denotes the complete set of parameters and $N$ the total number of parameters, the Neural Tangent Kernel (NTK) is defined as [12]*

$$\Theta_N(\boldsymbol{x}, \boldsymbol{y}; \boldsymbol{\theta}) = \sum_{n=1}^{N} \partial_{\theta_n} f(\boldsymbol{x}, \boldsymbol{\theta}) \partial_{\theta_n} f(\boldsymbol{y}, \boldsymbol{\theta}), \qquad (11)$$

*where $\partial_{\theta_n}$ denotes partial derivation w.r.t. the $n$-th parameter $\theta_n$.*

For one-hidden-layer networks with random, $\mathcal{O}(1)$-variance Gaussian initialisation of all the weights, and normalisation by $\sqrt{H}$ as in (3.1) and (3.2), the NTK converges to a deterministic limit $\Theta(\boldsymbol{x}, \boldsymbol{y})$ as $N \propto H \to \infty$ [12]. Furthermore, training $f(\boldsymbol{x}, \boldsymbol{\theta}) - f(\boldsymbol{x}, \boldsymbol{\theta}_0)$, with $\boldsymbol{\theta}_0$ denoting the network parameters at initialisation, under gradient descent on the mean squared error is equivalent to performing ridgeless regression with kernel $\Theta(\boldsymbol{x}, \boldsymbol{y})$ [12]. The following lemmas relate the NTK of convolutional and local architectures acting on $d$-dimensional inputs to that of a fully-connected architecture acting on $s$-dimensional inputs. Both lemmas are proved in Appendix B.

**Lemma 3.1.** *Call $\Theta^{FC}$ the NTK of a fully-connected network function acting on $s$-dimensional inputs and $\Theta^{CN}$ the NTK of a convolutional network function (3.1) with filter size $s$ acting on $d$-dimensional inputs. Then*

$$\Theta^{CN}(\boldsymbol{x}, \boldsymbol{y}) = \frac{1}{|\mathcal{P}|^2} \sum_{i,j \in \mathcal{P}} \Theta^{FC}(\boldsymbol{x}_i, \boldsymbol{y}_j) \tag{12}$$

As the functions in Eq. (2), $\Theta^{CN}$ is written as a combination of lower-dimensional constituent kernels $\Theta^{FC}$ acting on patches, and the dimensionality of the constituent kernel coincides with the filter size of the corresponding network. This observation extends to local kernels, via

**Lemma 3.2.** *Call $\Theta^{LC}$ the NTK of a locally-connected network function (3.2) with filter size $s$ acting on $d$-dimensional inputs. Then*

$$\Theta^{LC}(\boldsymbol{x}, \boldsymbol{y}) = \frac{1}{|\mathcal{P}|} \sum_{i \in \mathcal{P}} \Theta^{FC}(\boldsymbol{x}_i, \boldsymbol{y}_i) \tag{13}$$

Following the general structure of Eq. (12) and Eq. (13), we introduce convolutional ($K^{CN}$) and local ($K^{LC}$) student and teacher kernels, defined as sums of lower-dimensional constituent kernels $C$,

$$K^{CN}(\boldsymbol{x}, \boldsymbol{y}) = |\mathcal{P}|^{-2} \sum_{i,j \in \mathcal{P}} C(\boldsymbol{x}_i, \boldsymbol{y}_j), \tag{14a}$$

$$K^{LC}(\boldsymbol{x}, \boldsymbol{y}) = |\mathcal{P}|^{-1} \sum_{i \in \mathcal{P}} C(\boldsymbol{x}_i, \boldsymbol{y}_i). \tag{14b}$$

The kernels in Eq. (14) are characterised by the dimensionality of the constituent kernel $C$, or *filter size $s$* (for the student, or $t$ for the teacher) and the nonanalytic behaviour of $C$ when the two arguments approach, i.e. $C(\boldsymbol{x}_i, \boldsymbol{y}_j) \sim \|\boldsymbol{x}_i - \boldsymbol{y}_j\|^{\alpha_s}$ (for the student, or $\|\boldsymbol{x}_i - \boldsymbol{y}_j\|^{\alpha_t}$ for the teacher) plus analytic contributions, with $\alpha_{s/t} \neq 2m$ for $m \in \mathbb{N}$. Using the kernels in Eq. (14) as covariances allows us to generate random target functions with the desired degree of locality $t$ (as in Eq. (2)), which can also be invariant for shifts of the patches. Having a student kernel as in Eq. (14) results in an estimator $f$ also having the form displayed in Eq. (2), with a different filter size with respect to the target function. The $\alpha$'s control the smoothness of these functions as, if $\alpha > 2n \in \mathbb{N}$, then the functions are at least $n$ times differentiable in the mean-square sense.

A notable example of such constituent kernels is the NTK of ReLU networks $\Theta^{FC}$, which presents a cusp at the origin corresponding to $\alpha_s = 1$ [32]. In addition, in the $H \to \infty$ limit, a network initialised with random weights converges to a Gaussian process [33, 34, 35]. For networks with ReLU activations, the covariance kernel of such process has nonanalytic behaviour with $\alpha_t = 3$ [36].

## 3.1 Mercer's decomposition of local and convolutional kernels

We now turn to describing how the eigendecomposition of the constituent kernel $C$ induces an eigendecomposition of convolutional and local kernels. We work under the following assumptions,

    *i)* The constituent kernel $C(\boldsymbol{x}, \boldsymbol{y})$ on $\mathbb{R}^s \times \mathbb{R}^s$ admits the following Mercer's decomposition,

$$C(\boldsymbol{x}, \boldsymbol{y}) = \sum_{\rho=1}^{\infty} \lambda_\rho \phi_\rho(\boldsymbol{x}) \phi_\rho(\boldsymbol{y}), \tag{15}$$

    with (ordered) eigenvalues $\lambda_\rho$ and eigenfunctions $\phi_\rho$ such that, with $p^{(s)}(d^s x)$ denoting the $s$-dimensional patch measure, $\phi_1(\boldsymbol{x}) = 1 \; \forall \boldsymbol{x}$ and $\int p^{(s)}(d^s x) \phi_\rho(\boldsymbol{x}) = 0$ for all $\rho > 1$;

$ii$) Convolutional and local kernels from Eq. (14) have *nonoverlapping* patches, i.e. $d$ is an integer multiple of $s$ and $\mathcal{P} = \{1 + n \times s \,|\, n = 1, \ldots, d/s\}$ with $|\mathcal{P}| = d/s$;

$iii$) The $s$-dimensional marginals on patches of the $d$-dimensional input measure $p^{(d)}(d^d x)$ are all identical and equal to $p^{(s)}(d^s x)$.

We stress here that the request of nonoverlapping patches in assumption $ii$) can be relaxed at the price of further assumptions, i.e. $C(\boldsymbol{x}, \boldsymbol{y}) = \mathcal{C}(\boldsymbol{x} - \boldsymbol{y})$ and data distributed uniformly on the torus, so that $C$ is diagonalised in Fourier space. The resulting eigendecompositions are qualitatively similar to those described in this section (details in Appendix C). Let us also remark that assumptions $i$) and $iii$)—together with all the assumptions on the data distribution that might follow—are technical in nature and required only to carry out the Mercer's decomposition analytically. We believe that the main results of this paper hold under much more general conditions, namely the support of the distribution being truly $d$-dimensional—such that the distance between neighbouring points in a collection of $P$ data points scales as $P^{-1/d}$—and the distribution itself decaying rapidly away from the mean or having compact support. Our experiments, discussed in Section 5, support this hypothesis.

**Lemma 3.3** (Spectra of convolutional kernels)**.** *Let $K^{CN}$ be a convolutional kernel defined as in Eq. (14a), with a constituent kernel $C$ satisfying assumptions $i$), $ii$) and $iii$) above. Then $K^{CN}$ admits the following Mercer's decomposition,*

$$K^{CN}(\boldsymbol{x}, \boldsymbol{y}) = \sum_{\rho=1}^{\infty} \Lambda_\rho \Phi_\rho(\boldsymbol{x}) \overline{\Phi_\rho(\boldsymbol{y})}, \tag{16}$$

*with eigenvalues and eigenfunctions*

$$\Lambda_1 = \lambda_1, \ \Phi_1(\boldsymbol{x}) = 1; \ \Lambda_\rho = \frac{s}{d} \lambda_\rho, \ \Phi_\rho(\boldsymbol{x}) = \sqrt{\frac{s}{d}} \sum_{i \in \mathcal{P}} \phi_\rho(\boldsymbol{x}_i) \, for \, \rho > 1. \tag{17}$$

**Lemma 3.4** (Spectra of local kernels)**.** *Let $K^{LC}$ be a local kernel defined as in Eq. (14b), with a constituent kernel $C$ satisfying assumptions $i$), $ii$) and $iii$) above. Then $K^{LC}$ admits the following Mercer's decomposition,*

$$K^{LC}(\boldsymbol{x}, \boldsymbol{y}) = \Lambda_1 \Phi_1(\boldsymbol{x}) + \sum_{\rho>1}^{\infty} \sum_{i \in \mathcal{P}} \Lambda_{\rho,i} \Phi_{\rho,i}(\boldsymbol{x}) \overline{\Phi_{\rho,i}(\boldsymbol{y})}, \tag{18}$$

*with eigenvalues and eigenfunctions ($\forall i \in \mathcal{P}$)*

$$\Lambda_1 = \lambda_1, \ \Phi_1(\boldsymbol{x}) = 1; \ \Lambda_{\rho,i} = \frac{s}{d} \lambda_\rho, \ \Phi_{\rho,i}(\boldsymbol{x}) = \phi_\rho(\boldsymbol{x}_i) \, for \, \rho > 1. \tag{19}$$

Under assumptions $i$), $ii$) and $iii$) above, lemmas 3.3 and 3.4 follow from the definitions of convolutional and local kernels and the eigendecompositions of the constituents (see Appendix C for a proof of the lemmas and generalisation to kernels with overlapping patches). In the next section, we explore the consequences of these results for the asymptotics of learning curves.

## 4 Asymptotic learning curves for ridgeless regression

In what follows, we consider explicitly translationally-invariant constituent kernels $C(\boldsymbol{x}_i, \boldsymbol{x}_j) = \mathcal{C}(\boldsymbol{x}_i - \boldsymbol{x}_j)$ and a $d$-dimensional data distribution $p(d^d x)$ which is uniform on the torus, so that all lower-dimensional marginals are also uniform on lower-dimensional tori. Under these conditions, all results of Section 3 can be extended to kernels with overlapping patches ($\mathcal{P} = \{1, \ldots, d\}$), so that the main results of this paper apply to nonoverlapping as well as overlapping-patches kernels. Furthermore, Mercer's decomposition Eq. (15) can be written in Fourier space [37], with $s$-dimensional plane waves $\phi_{\boldsymbol{k}}^{(s)}(\boldsymbol{x}) = e^{i\boldsymbol{k}\cdot\boldsymbol{x}}$ as eigenfunctions and the eigenvalues coinciding with the Fourier transform of $\mathcal{C}$. Furthermore, for kernels with filter size $s$ (or $t$) and positive smoothness exponent $\alpha_s$ (or $\alpha_t$), the eigenvalues decay with a power $-(s + \alpha_s)$ (or $-(t + \alpha_t)$) of the modulus of the wavevector $k = \sqrt{\boldsymbol{k} \cdot \boldsymbol{k}}$ [38]. In this setting, we obtain our main result:

**Theorem 4.1.** *Let $K_T$ be a $d$-dimensional convolutional kernel with a translationally-invariant $t$-dimensional constituent and leading nonanalyticity at the origin controlled by the exponent $\alpha_t > 0$. Let $K_S$ be a $d$-dimensional convolutional or local student kernel with a translationally-invariant $s$-dimensional constituent, and with a nonanalyticity at the origin controlled by the exponent $\alpha_s > 0$. Assume, in addition, that if the kernels have overlapping patches then $s \geq t$, whereas if the kernels have nonoverlapping patches $s$ is an integer multiple of $t$; and that data are uniformly distributed on a $d$-dimensional torus. Then, the following asymptotic equivalence holds in the limit $P \to \infty$,*

$$\mathcal{B}(P) \sim P^{-\beta}, \quad \beta = \alpha_t/s.$$

Theorem 4.1, together with Eq. (8) and the additional assumption $\alpha_t \leq 2(\alpha_s + s)$, yields the following expression for the learning curves asymptotics,

$$\epsilon(P) \sim P^{-\beta}, \quad \beta = \alpha_t/s. \tag{20}$$

As $\beta$ is independent of the embedding dimension $d$, we conclude that the curse of dimensionality is beaten when a convolutional target is learnt with a convolutional or local kernel. In fact, Eq. (20) indicates that there is no asymptotic advantage in using a convolutional rather than local student when learning a convolutional task, confirming the picture that locality, not weight sharing, is the main source of the convolutional architecture's performances [6]. In Appendix D we show that the generalization error of a local student learning convolutional teacher decays as

$$\epsilon(P) \sim \left(\frac{P}{|\mathcal{P}|}\right)^{-\beta}, \quad \beta = \alpha_t/s. \tag{21}$$

Eq. (21) implies that including weight sharing only amounts to a rescaling of $P$ by a factor $|\mathcal{P}|$—the size of the translation group over patches—recovering the result obtained in [28]. Intuitively, a local student will need $|\mathcal{P}|$ times more points than a convolutional student to learn the target with comparable accuracy, since it has to learn the same local function in all the possible $|\mathcal{P}|$ locations. The predictions in Eq. (20) and Eq. (21) are confirmed empirically, as discussed in Section 5 and Appendix G. Let us mention in particular that, although our predictions are valid only asymptotically, they hold already in the range $P \sim 10^2 - 10^3$, consistently with the number of training points typically used in applications.

Theorem 4.1 is proven in Appendix D and extended to the case of a local teacher and local student in Appendix E. Here we sketch the proof for the nonoverlapping case, which begins with the calculation of the variance of the coefficients of the target function in the student basis. By indexing the coefficients with the $s$-dimensional wavevectors $\boldsymbol{k}$,

$$\begin{aligned}
\mathbb{E}[|c_{\boldsymbol{k}}|^2] &= \int_{[0,1]^d} d^d x \, \Phi_{\boldsymbol{k}}(\boldsymbol{x}) \int_{[0,1]^d} d^d y \, \overline{\Phi_{\boldsymbol{k}}}(\boldsymbol{y}) \mathbb{E}[f^*(\boldsymbol{x}) f^*(\boldsymbol{y})] \\
&= \int_{[0,1]^d} d^d x \, \Phi_{\boldsymbol{k}}(\boldsymbol{x}) \int_{[0,1]^d} d^d y \, \overline{\Phi_{\boldsymbol{k}}}(\boldsymbol{y}) K_T(\boldsymbol{x}, \boldsymbol{y}).
\end{aligned} \tag{22}$$

If the size of teacher and student coincide, $s = t$, teacher and student have the same eigenfunctions. Thus, using the eigenvalue equation Eq. (6) of the teacher yields $\mathbb{E}[|c_{\boldsymbol{k}}|^2] \sim k^{-(\alpha_t+t)} = k^{-(\alpha_t+s)}$. After ranking eigenvalues by $k$, with multiplicity $k^{s-1}$ from all the wavevectors having the same modulus $k$, one has

$$\mathcal{B}(P) = \sum_{\{\boldsymbol{k}|k>P^{1/s}\}} k^{-(\alpha_t+s)} \sim \int_{P^{1/s}}^{\infty} dk\, k^{s-1} k^{-(\alpha_t+s)} \sim P^{-\frac{\alpha_t}{s}}. \tag{23}$$

When the filter size of the teacher $t$ is lowered, some of the coefficients $\mathbb{E}[|c_{\boldsymbol{k}}|^2]$ vanish. As the target function becomes a composition of $t$-dimensional constituents, the only non-zero coefficients are found for $\boldsymbol{k}$'s which lie in some $t$-dimensional subspaces of the $s$-dimensional Fourier space. These subspaces correspond to the $\boldsymbol{k}$ having at most a patch of $t$ consecutive non-vanishing components. In other words, $\mathbb{E}[|c_{\boldsymbol{k}}|^2]$ is finite only if $\boldsymbol{k}$ is effectively $t$-dimensional and the integral on the right-hand side of Eq. (23) becomes $t$-dimensional, thus

$$\mathcal{B}(P) \sim \int_{P^{1/s}}^{\infty} dk\, k^{t-1} k^{-(\alpha_t+t)} \sim P^{-\frac{\alpha_t}{s}}. \tag{24}$$

If the teacher patches are not contained in the student ones, the target cannot be represented with a combination of student eigenfunctions, hence the error asymptotes to a finite value when $P \to \infty$.

# 5   Empirical learning curves for ridgeless regression

This section investigates numerically the asymptotic behaviour of the learning curves for our teacher-student framework. We consider different combinations of convolutional and local teachers and students with overlapping patches and Laplacian constituent kernels, i.e. $\mathcal{C}(\boldsymbol{x}_i - \boldsymbol{x}_j) = e^{-\|\boldsymbol{x}_i - \boldsymbol{x}_j\|}$. In order to test the robustness of our results to the data distribution, data are uniformly generated in the hypercube $[0,1]^d$ (results in Fig. 1) or on a $d$-hypersphere (results in Appendix G). Fig. 1 shows learning curves for both convolutional (left panels) and local (right panels) students learning a convolutional target function. The results in the case of a local teacher are presented in Appendix G, and display no qualitative differences.

In the following, we always refer to Fig. 1. Panels A and B show that, with $\alpha_t = \alpha_s = 1$, our prediction $\beta = 1/s$ holds independently of the embedding dimension $d$. Furthermore, notice that fixing the dimension $d$ and the teacher filter size $t$, the generalisation errors of a convolutional and a local student with the same filter size differ only by a multiplicative constant independent of $P$. Indeed, the shift-invariant nature of the convolutional student only results in a pre-asymptotic correction to our estimate of the generalisation error $\mathcal{B}(P)$. In Appendix G, we check that this multiplicative constant corresponds to rescaling $P$ by the number of patches, as predicted in Section 4. Panels C and D show learning curves for several values of $s$ and fixed $t$. The curse of dimensionality is recovered when the size of the student filters coincides with the input dimension, both for local and convolutional students. Finally, panels E and F show learning curves for fixed $t$ and $s$ being smaller than, equal to or larger than $t$. We stress that, when $s < t$ the student kernel cannot reproduce the target function, hence the error does not decrease by increasing $P$. Further details on the experiments are provided in Appendix G, together with learning curves for data distributed uniformly on the unit sphere $\mathbb{S}^{d-1}$ and for regression with the actual analytical and empirical NTKs of one-hidden-layer convolutional networks. It is worthwhile to notice that experiments are always in excellent agreement with our predictions, despite using data distributions that are out of the hypotheses of Theorem 4.1. Indeed, for regression with the actual NTK even the assumption of translationally-invariant constituents is violated. Moreover, we report the learning curves of local kernels on the CIFAR-10 dataset showing that smaller filter sizes correspond to faster decays even for real and anisotropic data distributions, in agreement with the picture emerging from our synthetic model.

# 6   Asymptotics of learning curves with decreasing ridge

We now prove an upper bound for the exponent $\beta$ implying that the curse of dimensionality is beaten by a local or convolutional kernel learning a convolutional target (as in Eq. (2)), using the framework developed in [21] and a natural universality assumption on the kernel eigenfunctions. It is worth noticing that this framework does not require the target function to be generated by a teacher kernel. Proofs are presented in Appendix F. Let $\mathcal{D}(\Lambda)$ denote the density of eigenvalues of the student kernel, $\mathcal{D}(\Lambda) = \sum_\rho \delta(\Lambda - \Lambda_\rho)$, with $\delta(x)$ denoting Dirac delta function. Having a random target function with coefficients $c_\rho$ in the kernel eigenbasis having variance $\mathbb{E}[|c_\rho|^2]$, one can define the following reduced density (with respect to the teacher):

$$\mathcal{D}_T(\Lambda) = \sum_{\{\rho|\mathbb{E}[|c_\rho|^2]>0\}} \delta(\Lambda - \Lambda_\rho) \tag{25}$$

$\mathcal{D}_T(\Lambda)$ counts eigenvalues for which the target has a non-zero variance, such that:

$$\sum_\rho \mathbb{E}[|c_\rho|^2] = \int d\Lambda \mathcal{D}_T(\Lambda) c^2(\Lambda), \tag{26}$$

where the function $c(\Lambda)$ is defined by $c^2(\Lambda_\rho) = \mathbb{E}[|c_\rho|^2]$ for all $\rho$ such that $\mathbb{E}[|c_\rho|^2] > 0$. The following theorem then follows from the results of [21].

**Theorem 6.1.** *Let us consider a positive-definite kernel $K$ with eigenvalues $\Lambda_\rho$, $\sum_\rho \Lambda_\rho < \infty$, and eigenfunctions $\Phi_\rho$ learning a (random) target function $f^*$ in kernel ridge regression (Eq. (3)) with ridge $\lambda$ from $P$ observations $f^*(\boldsymbol{x}^\mu)$, with $\boldsymbol{x}^\mu \in \mathbb{R}^d$ drawn from a certain probability distribution. Let us denote with $\mathcal{D}_T(\Lambda)$ the reduced density of kernel eigenvalues with respect to the target and $\epsilon(\lambda, P)$ the generalisation error and also assume that*

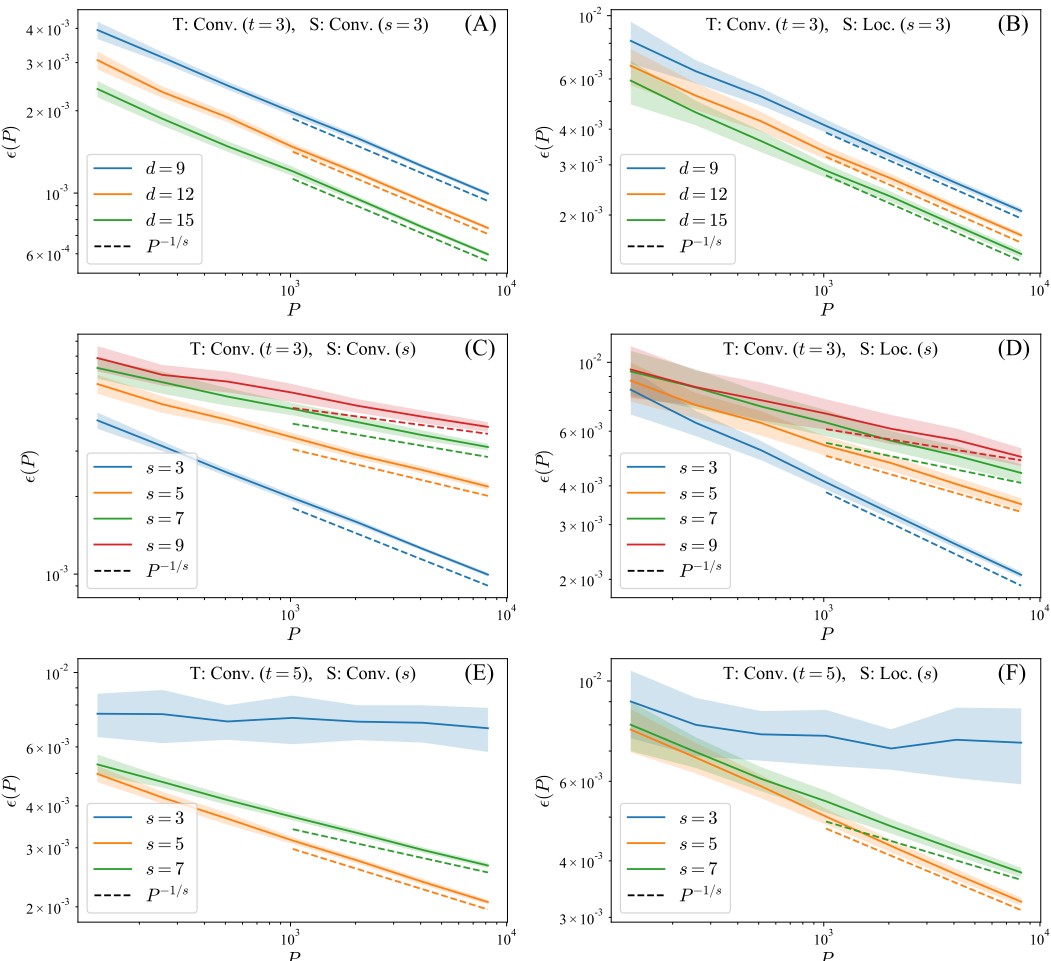

Figure 1: Learning curves for different combinations of convolutional teachers with convolutional (left panels) and local (right panels) students. The teacher and student filter sizes are denoted with $t$ and $s$ respectively. Data are sampled uniformly in the hypercube $[0, 1]^d$, with $d = 9$ if not specified otherwise. Solid lines are the results of numerical experiments averaged over 128 realisations and the shaded areas represent the empirical standard deviations. The predicted scalings are shown by dashed lines. All the panels are discussed in Section 5, while additional details on experiments are reported in Appendix G, together with additional experiments.

    $i)$ *For any $P$-tuple of indices $\rho_1, \ldots, \rho_P$, the vector $(\Phi_{\rho_1}(\boldsymbol{x}^1), \ldots, \Phi_{\rho_P}(\boldsymbol{x}^P))$ is a Gaussian random vector;*

    $ii)$ *The target function can be written in the kernel eigenbasis with coefficients $c_\rho$ and $c^2(\Lambda_\rho) = \mathbb{E}[|c_\rho|^2]$, with $\mathcal{D}_T(\Lambda) \sim \Lambda^{-(1+r)}$, $c^2(\Lambda) \sim \Lambda^q$ asymptotically for small $\Lambda$ and $r > 0$, $r < q < r + 2$;*

*Then the following equivalence holds in the joint $P \to \infty$ and $\lambda \to 0$ limit with $1/(\lambda\sqrt{P}) \to 0$:*

$$\epsilon(\lambda, P) \sim \sum_{\{\rho | \Lambda_\rho < \lambda\}} \mathbb{E}[|c_\rho|^2] = \int_0^\lambda d\Lambda \mathcal{D}_T(\Lambda) c^2(\Lambda). \tag{27}$$

Note that the assumption $i)$ of the theorem on the Gaussianity of the eigenbasis does not hold in our setup where the $\Phi_\rho$'s are plane waves. However, the random variables $\Phi_\rho(\boldsymbol{x}^\mu)$ have a probability density with compact support. It is thus natural to assume that a Gaussian universality assumption holds, i.e. that Theorem 6.1 applies to our problem. With this assumption, we obtain the following

**Corollary 6.1.1.** *Performing kernel ridge regression in a teacher-student scenario with smoothness exponents $\alpha_t$ (teacher) and $\alpha_s$ (student), with ridge $\lambda \sim P^{-\gamma}$ and $0 < \gamma < 1/2$, under the joint hypotheses of Theorem 4.1 and Theorem 6.1, the exponent governing the asymptotic scaling of the generalisation error with $P$ is given by:*

$$\beta = \gamma \frac{\alpha_t}{\alpha_s + s}, \tag{28}$$

which does not vanish in the limit $d \to \infty$. Furthermore, Eq. (28) depends on $s$ and not on $t$ as the prediction of Eq. (20).

# 7 Conclusions and future work

Our work shows that, even in large dimension $d$, a function can be learnt efficiently if it can be expressed as a sum of constituent functions each depending on a smaller number of variables $t$, by performing regression with a kernel that entails such a compositional structure with $s$-dimensional constituents. The learning curve exponent is then independent of $d$ and governed by $s$ if $s \geq t$, optimal for $s = t$ and null if $s < t$.

In the context of image classification, this result relates to the "Bag of Words" viewpoint. Consider for example two-dimensional images consisting of $M$ features of $t$ adjacent pixels, and that different classes correspond to distinct subsets of (possibly shared) features. If features can be located anywhere, then data lie on a $2M$-dimensional manifold. On the one hand, we expect a one-hidden-layer convolutional network with filter size $s \geq t$ to learn well with a learning curve exponent governed by $s$ and independent of $M$. On the other hand, a fully-connected network would suffer from the curse of dimensionality for large $M$.

Our work does not consider that the compositional structure of real data is hierarchical, with large features that consist of smaller sub-features. It is intuitively clear that depth and locality taken together are well-suited for such data structure [8, 6]. Extending the present teacher-student framework to this case would offer valuable quantitative insights into the question of how many data are required to learn such tasks.

# Acknowledgements

We thank Alberto Bietti, Stefano Spigler, Antonio Sclocchi, Leonardo Petrini, Mario Geiger, and Umberto Maria Tomasini for helpful discussions. This work was supported by a grant from the Simons Foundation (#454953 Matthieu Wyart).

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
