# Supplementary material for 'Locality defeats the curse of dimensionality in convolutional teacher-student scenarios'

**Alessandro Favero** [‡]
Institute of Physics
École Polytechnique Fédérale de Lausanne
`alessandro.favero@epfl.ch`

**Francesco Cagnetta** [‡]
Institute of Physics
École Polytechnique Fédérale de Lausanne
`francesco.cagnetta@epfl.ch`

**Matthieu Wyart**
Institute of Physics
École Polytechnique Fédérale de Lausanne
`matthieu.wyart@epfl.ch`

## Contents

## A   Spectral bias in kernel regression

In this appendix we provide additional details about the derivation of Eq. (8) within the framework of [17, 18]. Let us begin by recalling the definition of the kernel ridge regression estimator $f$ of a target function $f^*$,

$$f = \operatorname*{argmin}_{f \in \mathcal{H}} \left\{ \frac{1}{P} \sum_{\mu=1}^{P} (f(\boldsymbol{x}^\mu) - f^*(\boldsymbol{x}^\mu))^2 + \lambda \|f\|_{\mathcal{H}}^2 \right\}, \tag{S1}$$

---

[‡]Equal contribution.

35th Conference on Neural Information Processing Systems (NeurIPS 2021).

where $\mathcal{H}$ denotes the Reproducing Kernel Hilbert Space (RKHS) of the kernel $K(\boldsymbol{x}, \boldsymbol{y})$. After introducing the Mercer's decomposition of the kernel,

$$K(\boldsymbol{x}, \boldsymbol{y}) = \sum_{\rho=1}^{\infty} \lambda_\rho \phi_\rho(\boldsymbol{x}) \overline{\phi_\rho(\boldsymbol{y})}, \quad \int p\left(d^d y\right) K(\boldsymbol{x}, \boldsymbol{y}) \phi_\rho(\boldsymbol{y}) = \lambda_\rho \phi_\rho(\boldsymbol{x}). \tag{S2}$$

the RKHS can be characterised as a subset of the space of functions lying in the span of the kernel eigenbasis,

$$\mathcal{H} = \left\{ f = \sum_{\rho=1}^{\infty} a_\rho \phi_\rho(\boldsymbol{x}) \;\middle|\; \sum_{\rho=1}^{\infty} \frac{|a_\rho|^2}{\lambda_\rho} < \infty \right\}. \tag{S3}$$

In other words, the RKHS contains functions having a finite norm $||f||_{\mathcal{H}} = \sqrt{\langle f, f \rangle_{\mathcal{H}}}$ with respect to the following inner product,

$$f(\boldsymbol{x}) = \sum_\rho a_\rho \phi_\rho(\boldsymbol{x}), \;\; f'(\boldsymbol{x}) = \sum_\rho a'_\rho \phi_\rho(\boldsymbol{x}), \;\; \langle f, f' \rangle_{\mathcal{H}} = \sum_\rho \frac{a_\rho a'_\rho}{\lambda_\rho}. \tag{S4}$$

Given any target function $f^*$ lying in the span of the kernel eigenbasis, the mean squared generalisation error of the kernel ridge regression estimator reads

$$\epsilon(\lambda, \{\boldsymbol{x}^\mu\}) = \int p(d^d \boldsymbol{x}) \left(f(\boldsymbol{x}) - f^*(\boldsymbol{x})\right)^2 = \sum_{\rho=1}^{\infty} |a_\rho(\lambda, \{\boldsymbol{x}^\mu\}) - c_\rho|^2, \tag{S5}$$

with $c_\rho$ denoting the $\rho$-th coefficient of the target $f^*$ and $a_\rho$ that of the estimator $f$, which depends on the ridge $\lambda$ and on the training set $\{\boldsymbol{x}^\mu\}_{\mu=1,\dots,P}$. Notice that, as $f$ belongs to $\mathcal{H}$ by definition, $\sum_\rho |a_\rho|^2/\lambda_\rho < +\infty$, whereas the $c_\rho$'s are free to take any value.

The authors of [17, 18] found a heuristic expression for the average of the mean squared error over realisations of the training set $\{\boldsymbol{x}^\mu\}$. Such expression, based on the replica method of statistical physics, reads[1]

$$\epsilon(\lambda, P) = \partial_\lambda \left( \frac{\kappa_\lambda(P)}{P} \right) \sum_\rho \frac{\kappa_\lambda(P)^2}{(P\lambda_\rho + \kappa_\lambda(P))^2} |c_\rho|^2, \tag{S6}$$

where $\kappa(P)$ satisfies

$$\frac{\kappa_\lambda(P)}{P} = \lambda + \frac{1}{P} \sum_\rho \frac{\lambda_\rho \kappa_\lambda(P)/P}{\lambda_\rho + \kappa_\lambda(P)/P}. \tag{S7}$$

In short, the replica method works as follows [39]: first one defines an energy function $E(f)$ as the argument of the minimum in Eq. (S1), then attribute to the predictor $f$ a Boltzmann-like probability distribution $P(f) = Z^{-1} e^{-\beta E(f)}$, with $Z$ a normalisation constant and $\beta > 0$. As $\beta \to \infty$, the probability distribution $P(f)$ concentrates around the solution of the minimisation problem of Eq. (S1), i.e. the predictor of kernel regression. Hence, one can replace $f$ in the right-hand side of Eq. (S5) with an average over $P(f)$ at finite $\beta$, then perform the limit $\beta \to \infty$ after the calculation so as to recover the generalisation error. The simplification stems from the fact that, once $f$ is replaced with its eigendecomposition, the energy function $E(f)$ becomes a quadratic function of the coefficients $c_\rho$. Then, under the assumption that the data distribution enters only via the first and second moments of the eigenfunctions $\phi_\rho(\boldsymbol{x})$ w.r.t $\boldsymbol{x}$, all averages in Eq. (S5) reduce to Gaussian integrals.

Mathematically, $\kappa_\lambda(P)/P$ is related to the Stieltjes transform [40] of the Gram matrix $\mathbb{K}_P/P$ in the large-$P$ limit. Intuitively, it plays the role of a threshold: the modal contributions to the error tend to 0 for $\rho$ such that $\lambda_\rho \gg k_\lambda(P)/P$, and to $\mathbb{E}[|c_\rho|^2]$ for $\rho$ such that $\lambda_\rho \ll k_\lambda(P)/P$. This is equivalent to saying that the algorithm predictor $f(\boldsymbol{x})$ captures only the eigenmodes having eigenvalue larger than $k_\lambda(P)/P$ (see also [41, 21]).

This intuitive picture can actually be exploited in order to extract the learning curve exponent $\beta$ from the asymptotic behaviour of Eq. (S6) and Eq. (S7) in the ridgeless limit $\lambda \to 0^+$. In the following, we assume that both the kernel and the target function have a power-law spectrum, in particular

---

[1]Notice that the risk considered in [17, 18] slightly differs from Eq. (S1) by a factor $1/P$ in front of the sum.

$\lambda_\rho \sim \rho^{-a}$ and $\mathbb{E}[|c_\rho^*|^2] \sim \rho^{-b}$, with $2a > b - 1$. First, we approximate the sum over modes in Eq. (S7) with an integral using the Euler-Maclaurin formula. Then we substitute the eigenvalues inside the integral with their asymptotic limit, $\lambda_\rho = A\rho^{-a}$. Since, $\kappa_0(P)/P \to 0$ as $P \to \infty$, both these operations result in an error which is asymptotically independent of $P$. Namely,

$$
\begin{aligned}
\frac{\kappa_0(P)}{P} &= \frac{\kappa_0(P)}{P} \frac{1}{P} \left( \int_0^\infty \frac{d\rho\, A\rho^{-a}}{A\rho^{-a} + \kappa_0(P)/P} + \mathcal{O}(1) \right) \\
&= \frac{\kappa_0(P)}{P} \frac{1}{P} \left( \left( \frac{\kappa_0(P)}{P} \right)^{-\frac{1}{a}} \int_0^\infty \frac{d\sigma\, \sigma^{\frac{1}{a}-1} A^{\frac{1}{a}} a^{-1}}{1 + \sigma} + \mathcal{O}(1) \right),
\end{aligned}
\tag{S8}
$$

where in the second line, we changed the integration variable from $\rho$ to $\sigma = \kappa_0(P)\rho^a/(AP)$. Since the integral in $\sigma$ is finite and independent of $P$, we obtain that $\kappa_0(P)/P = \mathcal{O}(P^{-a})$. Similarly, we find that the mode-independent prefactor $\partial_\lambda (\kappa_\lambda(P)/P)|_{\lambda=0} = \mathcal{O}(1)$. As a result we are left with, modulo some $P$-independent prefactors,

$$
\epsilon(P) \sim \sum_\rho \frac{P^{-2a}}{(A\rho^{-a} + P^{-a})^2} \mathbb{E}[|c_\rho|^2].
\tag{S9}
$$

Following the intuitive argument about the thresholding role of $\kappa_0(P)/P \sim P^{-a}$, it is convenient to split the sum in Eq. (S10) into sectors where $\lambda_\rho \gg \kappa_0(P)/P$, $\lambda_\rho \sim \kappa_0(P)/P$ and $\lambda_\rho \ll \kappa_0(P)/P$, i.e.,

$$
\epsilon(P) \sim \sum_{\rho \ll P} \frac{P^{-2a}}{(A\rho^{-a})^2} \mathbb{E}[|c_\rho|^2] + \sum_{\rho \sim P} \frac{1}{2} \mathbb{E}[|c_\rho|^2] + \sum_{\rho \gg P} \mathbb{E}[|c_\rho|^2].
\tag{S10}
$$

Finally, Eq. (8) is obtained by noticing that, under our assumptions on the decay of $\mathbb{E}[|c_\rho|^2]$ with $\rho$, the contribution of the sum over $\rho \ll P$ is subleading in $P$ whereas the other two sums can be gathered together.

## B  NTKs of convolutional and locally-connected networks

We begin this section by reviewing the computation of the NTK of a one-hidden-layer fully-connected network [16].

**Definition B.1** (one-hidden-layer FCN). *A one-hidden-layer fully-connected network with $H$ hidden neurons is defined as follows,*

$$
f^{FCN}(\boldsymbol{x}) = \frac{1}{\sqrt{H}} \sum_{h=1}^H a_h \sigma(\boldsymbol{w}_h \cdot \boldsymbol{x} + b_h),
\tag{S11}
$$

*where $\boldsymbol{x} \in \mathbb{R}^d$ is the input, $H$ is the width, $\sigma$ is a nonlinear activation function, $\{\boldsymbol{w}_h \in \mathbb{R}^d\}_{h=1}^H$, $\{b_h \in \mathbb{R}\}_{h=1}^H$, and $\{a_h \in \mathbb{R}\}_{h=1}^H$ are the network's parameters. The dot $\cdot$ denotes the standard Euclidean scalar product.*

Inserting Eq. (S11) into Eq. (11), one obtains

$$
\begin{aligned}
\Theta_N^{FC}(\boldsymbol{x}, \boldsymbol{y}; \boldsymbol{\theta}) = \frac{1}{H} \sum_{h=1}^H \big( &\sigma(\boldsymbol{w}_h \cdot \boldsymbol{x} + b_h)\sigma(\boldsymbol{w}_h \cdot \boldsymbol{y} + b_h) \\
&+ a_h^2 \sigma'(\boldsymbol{w}_h \cdot \boldsymbol{x} + b_h)\sigma'(\boldsymbol{w}_h \cdot \boldsymbol{y} + b_h)(\boldsymbol{x} \cdot \boldsymbol{y} + 1) \big),
\end{aligned}
\tag{S12}
$$

where $\sigma'$ denotes the derivative of $\sigma$ with respect to its argument. If all the parameters are initialised independently from a standard Normal distribution, $\Theta_N^{FC}(\boldsymbol{x}, \boldsymbol{y}; \boldsymbol{\theta})$ is a random-feature kernel that in the $H \to \infty$ limit converges to

$$
\begin{aligned}
\Theta^{FC}(\boldsymbol{x}, \boldsymbol{y}) = &\mathbb{E}_{\boldsymbol{w},b}[\sigma(\boldsymbol{w} \cdot \boldsymbol{x} + b)\sigma(\boldsymbol{w} \cdot \boldsymbol{y} + b)] \\
&+ \mathbb{E}_a[a^2]\mathbb{E}_{\boldsymbol{w},b}[\sigma'(\boldsymbol{w} \cdot \boldsymbol{x} + b)\sigma'(\boldsymbol{w} \cdot \boldsymbol{y} + b)](\boldsymbol{x} \cdot \boldsymbol{y} + 1).
\end{aligned}
\tag{S13}
$$

When $\sigma$ is the ReLU activation function, the expectations can be computed exactly using techniques from the literature of arc-cosine kernels [36]

$$\Theta^{FC}(\boldsymbol{x}, \boldsymbol{y}) = \frac{1}{2\pi} \sqrt{\|\boldsymbol{x}\|^2 + 1} \sqrt{\|\boldsymbol{y}\|^2 + 1} \left( \sin\varphi + (\pi - \varphi) \cos\varphi \right) \tag{S14}$$
$$+ \frac{1}{2\pi} (\boldsymbol{x} \cdot \boldsymbol{y} + 1)(\pi - \varphi),$$

with $\varphi$ denoting the angle

$$\varphi = \arccos\left( \frac{\boldsymbol{x} \cdot \boldsymbol{y} + 1}{\sqrt{\|\boldsymbol{x}\|^2 + 1} \sqrt{\|\boldsymbol{y}\|^2 + 1}} \right). \tag{S15}$$

Notice that, as commented in Section 3, for ReLU networks $\Theta^{FC}(\boldsymbol{x}, \boldsymbol{y})$ displays a cusp at $\boldsymbol{x} = \boldsymbol{y}$.

**Proof of Lemma 3.1**

*Proof.* Inserting Eq. (9) into Eq. (11),

$$\Theta_N^{CN}(\boldsymbol{x}, \boldsymbol{y}; \boldsymbol{\theta}) = \frac{1}{|\mathcal{P}|^2} \sum_{i,j \in \mathcal{P}} \left( \frac{1}{H} \sum_{h=1}^{H} \left( \sigma(\boldsymbol{w}_h \cdot \boldsymbol{x}_i + b_h) \sigma(\boldsymbol{w}_h \cdot \boldsymbol{y}_j + b_h) \right. \right. \tag{S16}$$
$$\left. \left. + a_h^2 \sigma'(\boldsymbol{w}_h \cdot \boldsymbol{x}_i + b_h) \sigma'(\boldsymbol{w}_h \cdot \boldsymbol{y}_j + b_h)(\boldsymbol{x}_i \cdot \boldsymbol{y}_j + 1) \right) \right)$$

In the previous line, the single terms of the summation over patches are the random-feature kernels $\Theta_N^{FC}$ obtained in Eq. (S12) acting on $s$-dimensional inputs, i.e. the patches of $\boldsymbol{x}$ and $\boldsymbol{y}$. Therefore,

$$\Theta_N^{CN}(\boldsymbol{x}, \boldsymbol{y}; \boldsymbol{\theta}) = \frac{1}{|\mathcal{P}|^2} \sum_{i,j \in \mathcal{P}} \Theta_N^{(FC)}(\boldsymbol{x}, \boldsymbol{y}). \tag{S17}$$

If all the parameters are initialised independently from a standard Normal distribution, the $H \to \infty$ limit of Eq. (S17) yields Eq. (12).

∎

**Proof of Lemma 3.2**

*Proof.* Inserting Eq. (10) into Eq. (11),

$$\Theta_N^{LC}(\boldsymbol{x}, \boldsymbol{y}; \boldsymbol{\theta}) = \frac{1}{|\mathcal{P}|} \sum_{i \in \mathcal{P}} \left( \frac{1}{H} \sum_{h=1}^{H} \left( \sigma(\boldsymbol{w}_{h,i} \cdot \boldsymbol{x}_i + b_{h,i}) \sigma(\boldsymbol{w}_{h,i} \cdot \boldsymbol{y}_i + b_{h,i}) \right. \right. \tag{S18}$$
$$\left. \left. + a_{h,i}^2 \sigma'(\boldsymbol{w}_{h,i} \cdot \boldsymbol{x}_i + b_{h,i}) \sigma'(\boldsymbol{w}_{h,i} \cdot \boldsymbol{y}_i + b_{h,i})(\boldsymbol{x}_i \cdot \boldsymbol{y}_i + 1) \right) \right).$$

In the previous line, the single terms of the summation over patches are the random-feature kernels $\Theta_N^{FC}$ obtained in Eq. (S12) acting on $s$-dimensional inputs, i.e. the patches of $\boldsymbol{x}$ and $\boldsymbol{y}$. Therefore,

$$\Theta_N^{LC}(\boldsymbol{x}, \boldsymbol{y}; \boldsymbol{\theta}) = \frac{1}{|\mathcal{P}|} \sum_{i \in \mathcal{P}} \Theta_N^{(FC)}(\boldsymbol{x}_i, \boldsymbol{y}_i). \tag{S19}$$

If all the parameters are initialised independently from a standard Normal distribution, Eq. (13) is recovered in the $H \to \infty$ limit.

∎

# C Mercer's decomposition of convolutional and local kernels

In this section we prove the eigendecompositions introduced in Lemma 3.3 and Lemma 3.4, then extend them to overlapping-patches kernel (cf. C.1). We define the scalar product in input space between two (complex) functions $f$ and $g$ as

$$\langle f, g \rangle = \int p(d^d x)\, f(\boldsymbol{x})\overline{g(\boldsymbol{x})}. \tag{S20}$$

**Proof of Lemma 3.3**

*Proof.* We start by proving orthonormality of the eigenfunctions. By writing the $d$-dimensional eigenfunctions $\Phi_\rho$ in terms of the $s$-dimensional eigenfunctions $\phi_\rho$ of the constituent kernel as in Eq. (17), for $\rho, \sigma \neq 1$,

$$\langle \Phi_\rho, \Phi_\sigma \rangle = \frac{s}{d} \sum_{i,j \in \mathcal{P}} \int p(d^d x)\phi_\rho(\boldsymbol{x}_i)\overline{\phi_\sigma(\boldsymbol{x}_j)}. \tag{S21}$$

Separating the term in the sum over patches in which $i$ and $j$ coincide from the others, and since the patches are not overlapping, the RHS can be written as

$$\frac{s}{d} \sum_{i \in \mathcal{P}} \int p(d^s x_i)\phi_\rho(\boldsymbol{x}_i)\overline{\phi_\sigma(\boldsymbol{x}_i)} + \sum_{i,j \neq i \in \mathcal{P}} \int p(d^s x_i)\phi_\rho(\boldsymbol{x}_i) \int p(d^s x_j)\overline{\phi_\sigma(\boldsymbol{x}_j)}. \tag{S22}$$

From the orthonormality of the eigenfunctions $\phi_\rho$, the first integral is non-zero and equal to one only when $\rho = \sigma$, while, from assumption *i)*, $\int p^{(s)}(d^s x)\phi_\rho(\boldsymbol{x}) = 0$ for all $\rho > 1$, so that the second integral is always zero. Therefore,

$$\langle \Phi_\rho, \Phi_\sigma \rangle = \delta_{\rho,\sigma}, \text{ for } \rho, \sigma > 1. \tag{S23}$$

When $\rho = 1$ and $\sigma \neq 1$, $\int p(d^d x)\Phi_1(\boldsymbol{x})\overline{\Phi_\sigma(\boldsymbol{x})} = 0$ from assumption *i)*, i.e. $\Phi_1 = 1$ and $\int p^{(s)}(d^s x)\phi_\rho(\boldsymbol{x}) = 0$ for all $\rho > 1$. Finally, if $\rho = \sigma = 1$, $\int p(d^d x)\Phi_1(\boldsymbol{x})\overline{\Phi_1(\boldsymbol{x})} = 1$ trivially.

Then, we prove that the eigenfunctions and the eigenvalues defined in Eq. (17) satisfy the kernel eigenproblem. For $\rho = 1$,

$$\int p(d^d y)K^{CN}(\boldsymbol{x}, \boldsymbol{y}) = \int p(d^d y)\frac{s^2}{d^2} \sum_{i,j \in \mathcal{P}} C(\boldsymbol{x}_i, \boldsymbol{y}_j) = \frac{s^2}{d^2} \sum_{i,j \in \mathcal{P}} \lambda_1 = \Lambda_1, \tag{S24}$$

where we used $\int p^{(s)}(d^s y)C(\boldsymbol{x}, \boldsymbol{y}) = \lambda_1$ from assumption *i)*. For $\rho > 1$,

$$\int p(d^d y)K^{CN}(\boldsymbol{x}, \boldsymbol{y})\Phi_\rho(\boldsymbol{y}) = \int p(d^d y)\frac{s^2}{d^2} \sum_{i,j \in \mathcal{P}} C(\boldsymbol{x}_i, \boldsymbol{y}_j)\sqrt{\frac{s}{d}}\sum_{l \in \mathcal{P}}\phi_\rho(\boldsymbol{y}_l). \tag{S25}$$

Splitting the sum over $l$ into the term with $l = j$ and the remaining ones, the RHS can be written as

$$\frac{s^2}{d^2} \sum_{i,j \in \mathcal{P}} \left( \int p(d^s y_j)C(\boldsymbol{x}_i, \boldsymbol{y}_j)\sqrt{\frac{s}{d}}\phi_\rho(\boldsymbol{y}_j) \right. \tag{S26}$$

$$\left. + \int p(d^s y_j)C(\boldsymbol{x}_i, \boldsymbol{y}_j)\sqrt{\frac{s}{d}} \sum_{l \neq j \in \mathcal{P}} \int p(d^s y_l)\phi_\rho(\boldsymbol{y}_l) \right).$$

Using assumption *i)*, the third integral is always zero, therefore

$$\int p(d^d y) K^{CN}(\boldsymbol{x}, \boldsymbol{y}) \Phi_\rho(\boldsymbol{y}) = \frac{s^2}{d^2} \sum_{i,j \in \mathcal{P}} \lambda_\rho \sqrt{\frac{s}{d}} \phi_\rho(\boldsymbol{x}_i) = \Lambda_\rho \Phi_\rho(\boldsymbol{x}). \tag{S27}$$

Finally, we prove the expansion of Eq. (16) from the definition of $K^{CN}$,

$$K^{CN}(\boldsymbol{x}, \boldsymbol{y}) = \frac{s^2}{d^2} \sum_{i,j \in \mathcal{P}} C(\boldsymbol{x}_i, \boldsymbol{y}_j) \tag{S28}$$

$$= \frac{s^2}{d^2} \sum_{i,j \in \mathcal{P}} \sum_\rho \lambda_\rho \phi_\rho(\boldsymbol{x}_i) \overline{\phi_\rho(\boldsymbol{y}_j)}$$

$$= \lambda_1 \frac{s^2}{d^2} \sum_{i,j \in \mathcal{P}} \phi_1(\boldsymbol{x}_i) \overline{\phi_1(\boldsymbol{y}_j)} + \sum_{\rho > 1} \left( \frac{s}{d} \lambda_\rho \right) \left( \sqrt{\frac{s}{d}} \sum_{i \in \mathcal{P}} \phi_\rho(\boldsymbol{x}_i) \right) \left( \sqrt{\frac{s}{d}} \sum_{j \in \mathcal{P}} \overline{\phi_\rho(\boldsymbol{y}_j)} \right)$$

$$= \sum_\rho \Lambda_\rho \Phi_\rho(\boldsymbol{x}) \overline{\Phi_\rho(\boldsymbol{y})}.$$

∎

**Proof of Lemma 3.4**

*Proof.* We start again by proving the orthonormality of the eigenfunctions. By writing the $d$-dimensional eigenfunctions $\Phi_{\rho,i}$ in terms of the $s$-dimensional eigenfunctions $\phi_\rho$ of the constituent kernel as in Eq. (19), for $\rho, \sigma \neq 1$,

$$\langle \Phi_{\rho,i}, \Phi_{\sigma,j} \rangle = \int p(d^d x) \phi_\rho(\boldsymbol{x}_i) \overline{\phi_\sigma(\boldsymbol{x}_j)} = \delta_{\rho,\sigma} \delta_{i,j}, \tag{S29}$$

from the orthonormality of the eigenfunctions $\phi_\rho$ when $i = j$, and assumption *i)*, $\int p^{(s)}(d^s x) \phi_\rho(\boldsymbol{x}) = 0$ for all $\rho > 1$, when $i \neq j$. Moreover, as $\Phi_1(\boldsymbol{x}) = 1$, $\int p(d^d x) \Phi_1(\boldsymbol{x}) \overline{\Phi_{\sigma \neq 1,j}(\boldsymbol{x})} = 0$ and $\int p(d^d x) \Phi_1(\boldsymbol{x}) \overline{\Phi_1(\boldsymbol{x})} = 1$.

Then, we prove that the eigenfunctions and the eigenvalues defined in Eq. (19) satisfy the kernel eigenproblem. For $\rho = 1$,

$$\int p(d^d y) K^{LC}(\boldsymbol{x}, \boldsymbol{y}) = \int p(d^d y) \frac{s}{d} \sum_{i \in \mathcal{P}} C(\boldsymbol{x}_i, \boldsymbol{y}_i) = \frac{s}{d} \sum_{i \in \mathcal{P}} \lambda_1 = \Lambda_1, \tag{S30}$$

where we used $\int p^{(s)}(d^s y) C(\boldsymbol{x}, \boldsymbol{y}) = \lambda_1$ from assumption *i)*. For $\rho > 1$,

$$\int p(d^d y) K^{LC}(\boldsymbol{x}, \boldsymbol{y}) \Phi_{\rho,i}(\boldsymbol{y}) = \int p(d^d y) \frac{s}{d} \sum_{j \in \mathcal{P}} C(\boldsymbol{x}_j, \boldsymbol{y}_j) \phi_\rho(\boldsymbol{y}_i). \tag{S31}$$

Splitting the sum over $j$ in the term for which $j = i$ and the remaining ones, the RHS can be written as

$$\frac{s}{d} \int p(d^s y_i) C(\boldsymbol{x}_i, \boldsymbol{y}_i) \phi_\rho(\boldsymbol{y}_i) + \frac{s}{d} \sum_{j \neq i \in \mathcal{P}} \int p(d^s y_j) C(\boldsymbol{x}_j, \boldsymbol{y}_j) \int p(d^s y_i) \phi_\rho(\boldsymbol{y}_i). \tag{S32}$$

Using assumption *i)*, the third integral is always zero, therefore

$$\int p(d^d y) K^{CN}(\boldsymbol{x}, \boldsymbol{y}) \Phi_\rho(\boldsymbol{y}) = \frac{s}{d} \lambda_\rho \phi_\rho(\boldsymbol{x}_i) = \Lambda_{\rho,i} \Phi_{\rho,i}(\boldsymbol{x}). \tag{S33}$$

Finally, we prove the expansion of Eq. (16) from the definition of $K^{LC}$,

$$K^{LC}(\boldsymbol{x}, \boldsymbol{y}) = \frac{s}{d} \sum_{i \in \mathcal{P}} C(\boldsymbol{x}_i, \boldsymbol{y}_i) \tag{S34}$$

$$= \frac{s^2}{d^2} \sum_{i \in \mathcal{P}} \sum_{\rho} \lambda_\rho \phi_\rho(\boldsymbol{x}_i) \overline{\phi_\rho(\boldsymbol{y}_i)} \tag{S35}$$

$$= \lambda_1 \frac{s}{d} \sum_{i \in \mathcal{P}} \phi_1(\boldsymbol{x}_i) \overline{\phi_1(\boldsymbol{y}_i)} + \sum_{\rho > 1} \sum_{i \in \mathcal{P}} \left(\frac{s}{d} \lambda_\rho\right) \phi_\rho(\boldsymbol{x}_i) \overline{\phi_\rho(\boldsymbol{y}_i)} \tag{S36}$$

$$= \Lambda_1 \Phi_1(\boldsymbol{x}) \overline{\Phi_1(\boldsymbol{y})} + \sum_{\rho > 1} \sum_{i \in \mathcal{P}} \Lambda_{\rho,i} \Phi_{\rho,i}(\boldsymbol{x}) \overline{\Phi_{\rho,i}(\boldsymbol{y})}. \tag{S37}$$

∎

### C.1 Spectra of convolutional kernels with overlapping patches

In this section Lemma 3.3 and Lemma 3.4 are extended to kernels with overlapping patches, having $\mathcal{P} = \{1, \dots, d\}$ and $|\mathcal{P}| = d$. Such extension requires additional assumptions, which are stated below:

 i) The $d$-dimensional input measure $p^{(d)}(d^d x)$ is uniform on the $d$-torus $[0, 1]^d$;

 ii) The constituent kernel $C(\boldsymbol{x}, \boldsymbol{y})$ is translationally-invariant, isotropic and periodic,

$$C(\boldsymbol{x}, \boldsymbol{y}) = \mathcal{C}(||\boldsymbol{x} - \boldsymbol{y}||), \quad \mathcal{C}(||\boldsymbol{x} - \boldsymbol{y} + \boldsymbol{n}||) = \mathcal{C}(||\boldsymbol{x} - \boldsymbol{y}||) \quad \forall \boldsymbol{n} \in \mathbb{Z}^s. \tag{S38}$$

Assumptions $i)$ and $ii)$ above imply that $C(\boldsymbol{x}, \boldsymbol{y})$ can be diagonalised in Fourier space, i.e. (with $\boldsymbol{k}$ denoting the $s$-dimensional wavevector)

$$\mathcal{C}(\boldsymbol{x} - \boldsymbol{y}) = \sum_{\{\boldsymbol{k} = 2\pi\boldsymbol{n} | \boldsymbol{n} \in \mathbb{Z}^s\}} \lambda_{\boldsymbol{k}} \phi_{\boldsymbol{k}}(\boldsymbol{x}) \overline{\phi_{\boldsymbol{k}}(\boldsymbol{y})} = \sum_{\{\boldsymbol{k} = 2\pi\boldsymbol{n} | \boldsymbol{n} \in \mathbb{Z}^s\}} \lambda_{\boldsymbol{k}} e^{i\boldsymbol{k} \cdot (\boldsymbol{x} - \boldsymbol{y})}, \tag{S39}$$

and the eigenvalues $\lambda_{\boldsymbol{k}}$ depend only on the modulus of $\boldsymbol{k}$, $k = \sqrt{\boldsymbol{k} \cdot \boldsymbol{k}}$.

Let us introduce the following definitions, after recalling that a $s$-dimensional patch $\boldsymbol{x}_i$ of $\boldsymbol{x}$ is a contiguous subsequence of $\boldsymbol{x}$ starting at $x_i$, i.e.

$$\boldsymbol{x} = (x_1, x_2, \dots, x_d) \Rightarrow \boldsymbol{x}_i = (x_i, x_{i+1}, \dots, x_{i+s-1}), \tag{S40}$$

and that inputs are 'wrapped', i.e. we identify $x_{i+nd}$ with $x_i$ for all $n \in \mathbb{Z}$.

- Two patches $\boldsymbol{x}_i$ and $\boldsymbol{x}_j$ *overlap* if $\boldsymbol{x}_i \cap \boldsymbol{x}_j \neq \emptyset$. The overlap $\boldsymbol{x}_{i \cap j} \equiv \boldsymbol{x}_i \cap \boldsymbol{x}_j$ is an $o$-dimensional patch of $\boldsymbol{x}$, with $o = |\boldsymbol{x}_i \cap \boldsymbol{x}_j|$;
- let $\mathcal{P}$ denote the set of patch indices associated with a given kernel/architecture. We denote with $\mathcal{P}_i$ the set of indices of patches which overlap with $\boldsymbol{x}_i$, i.e. $\mathcal{P}_i = \{i - s + 1, \dots, i, \dots, i + s - 1\} = \{\mathcal{P}_{-,i}, i, \mathcal{P}_{+,i}\}$;
- Given two overlapping patches $\boldsymbol{x}_i$ and $\boldsymbol{x}_j$ with $o$-dimensional overlap, the union $\boldsymbol{x}_{i \cup j} \equiv \boldsymbol{x}_i \cup \boldsymbol{x}_j$ and differences $\boldsymbol{x}_{i \setminus j} \equiv \boldsymbol{x}_i \setminus \boldsymbol{x}_j$ and $\boldsymbol{x}_{j \setminus i} \equiv \boldsymbol{x}_j \setminus \boldsymbol{x}_i$ are all patches of $\boldsymbol{x}$, with dimensions $2s - o$, $s - o$ and $s - o$, respectively.

We also use the following notation for denoting subspaces of the $\boldsymbol{k}$-space $\cong \mathbb{Z}^s$,

$$\mathcal{F}^u = \{\boldsymbol{k} = 2\pi\boldsymbol{n} \,|\, \boldsymbol{n} \in \mathbb{Z}^s; \, n_1, n_u \neq 0; \, n_v = 0 \, \forall v \text{ s. t. } u < v \leq s\}. \tag{S41}$$

$\mathcal{F}^s$ is the set of all wavevectors $\boldsymbol{k}$ having nonvanishing extremal components $k_1$ and $k_s$. For $u < s$, $\mathcal{F}^u$ is formed by first considering only wavevectors having the last $s - u$ components equal to zero, then asking the resulting $u$-dimensional wavevectors to have nonvanishing extremal components. Practically, $\mathcal{F}^u$ contains wavevectors which can be entirely specified by the first $u$-dimensional patch $\boldsymbol{k}_1^{(u)} = (k_1, \dots, k_u)$ but not by the first $(u - 1)$-dimensional one. Notice that, in order to safely compare $\boldsymbol{k}$'s in different $\mathcal{F}$'s, we have introduced an apex $u$ denoting the dimensionality of the patch.

**Lemma C.1** (Spectra of overlapping convolutional kernels). *Let $K^{CN}$ be a convolutional kernel defined as in Eq. (14a), with $\mathcal{P} = \{1, \ldots, d\}$ and constituent kernel $C$ satisfying assumptions i), ii) above. Then, $K^{CN}$ admits the following Mercer's decomposition,*

$$K^{CN}(\boldsymbol{x}, \boldsymbol{y}) = \Lambda_{\boldsymbol{0}} + \sum_{u=1}^{s} \left( \sum_{\boldsymbol{k} \in \mathcal{F}^u} \Lambda_{\boldsymbol{k}} \Phi_{\boldsymbol{k}}(\boldsymbol{x}) \Phi_{\boldsymbol{k}}(\boldsymbol{y}) \right), \tag{S42}$$

*with eigenfunctions*

$$\Phi_{\boldsymbol{0}}(\boldsymbol{x}) = 1, \quad \Phi_{\boldsymbol{k}}(\boldsymbol{x}) = \frac{1}{\sqrt{d}} \sum_{i=1}^{d} \phi_{\boldsymbol{k}}(\boldsymbol{x}_i) \quad \forall \, \boldsymbol{k} \neq \boldsymbol{0}, \tag{S43}$$

*and eigenvalues*

$$\Lambda_{\boldsymbol{0}} = \lambda_{\boldsymbol{0}}, \quad \Lambda_{\boldsymbol{k}} = \frac{s - u + 1}{d} \lambda_{\boldsymbol{k}} \quad \forall \, \boldsymbol{k} \in \mathcal{F}^u \text{ with } u \leq s. \tag{S44}$$

*Proof.* We start by proving the orthonormality of the eigenfunctions. In general, by orthonormality of the $s$-dimensional plane waves $\phi_{\boldsymbol{k}}(\boldsymbol{x})$, we have

$$\langle \Phi_{\boldsymbol{k}}, \Phi_{\boldsymbol{q}} \rangle = \frac{1}{d} \int_{[0,1]^d} d^d x \left( \sum_{i=1}^{d} \phi_{\boldsymbol{k}}(\boldsymbol{x}_i) \right) \overline{\left( \sum_{j=1}^{d} \phi_{\boldsymbol{q}}(\boldsymbol{x}_j) \right)}$$

$$= \frac{1}{d} \sum_{i \in \mathcal{P}} \sum_{j \notin \mathcal{P}_i} \int d^s x_i \, e^{i \boldsymbol{k} \cdot \boldsymbol{x}_i} \int d^s x_j \, e^{-i \boldsymbol{q} \cdot \boldsymbol{x}_j} + \frac{1}{d} \sum_{i \in \mathcal{P}} \int d^s x_i \, e^{i(\boldsymbol{k} - \boldsymbol{q}) \cdot \boldsymbol{x}_i}$$

$$+ \frac{1}{d} \sum_{i \in \mathcal{P}} \sum_{j \in \mathcal{P}_{i,+}} \int (d^{s-o} x_{i \smallsetminus j}) \, e^{i \boldsymbol{k}_1^{(s-o)} \cdot \boldsymbol{x}_{i \smallsetminus j}} \int (d^o x_{i \cup j}) \, e^{i(\boldsymbol{k}_{s-o+1}^{(o)} - \boldsymbol{q}_1^{(o)}) \cdot \boldsymbol{x}_{i \cup j}} \int (d^{s-o} x_{j \smallsetminus i}) \, e^{i \boldsymbol{q}_{o+1}^{(s-o)} \cdot \boldsymbol{x}_{j \smallsetminus i}}$$

$$+ \frac{1}{d} \sum_{i \in \mathcal{P}} \sum_{j \in \mathcal{P}_{i,-}} \{ i \leftrightarrow j, \boldsymbol{k} \leftrightarrow \boldsymbol{q} \}$$

$$= \frac{1}{d} \sum_{i \in \mathcal{P}} \delta(\boldsymbol{k}, \boldsymbol{0}) \sum_{j \notin \mathcal{P}_i} \delta(\boldsymbol{q}, \boldsymbol{0}) + \frac{1}{d} \sum_{i \in \mathcal{P}} \delta(\boldsymbol{k}, \boldsymbol{q})$$

$$+ \frac{1}{d} \sum_{i \in \mathcal{P}} \left( \sum_{j \in \mathcal{P}_{i,+}} \delta(\boldsymbol{k}_1^{(s-o)}, \boldsymbol{0}) \, \delta(\boldsymbol{k}_{s-o+1}^{(o)}, \boldsymbol{q}_1^{(o)}) \, \delta(\boldsymbol{q}_{o+1}^{(s-o)}, \boldsymbol{0}) \right.$$

$$\left. + \sum_{j \in \mathcal{P}_{i,-}} \delta(\boldsymbol{q}_1^{(s-o)}, \boldsymbol{0}) \, \delta(\boldsymbol{k}_1^{(o)}, \boldsymbol{q}_{s-o+1}^{(o)}) \, \delta(\boldsymbol{k}_{o+1}^{(s-o)}, \boldsymbol{0}) \right), \tag{S45}$$

with $\delta(\boldsymbol{k}, \boldsymbol{q})$ denoting the multidimensional Kronecker delta. For fixed $i$, the three terms on the RHS correspond to $j$'s such that $\boldsymbol{x}_j$ does not overlap with $\boldsymbol{x}_i$, to $j = i$ and to $j$'s such that $\boldsymbol{x}_j$ overlaps with $\boldsymbol{x}_i$, respectively. We recall that, in patch notation, $\boldsymbol{k}_1^{(s-o)}$ denotes the subsequence of $\boldsymbol{k}$ formed with the first $s - o$ components and $\boldsymbol{k}_{s-o+1}^{(o)}$ the subsequence formed with the last $o$ components.

By taking $\boldsymbol{k}$ and $\boldsymbol{q}$ in $\mathcal{F}^s$, as $k_1, k_s \neq 0$ and $q_1, q_s \neq 0$, Eq. (S45) implies

$$\langle \Phi_{\boldsymbol{k}}, \Phi_{\boldsymbol{q}} \rangle = \delta(\boldsymbol{k}, \boldsymbol{q}). \tag{S46}$$

In addition, by taking $\boldsymbol{k} \in \mathcal{F}^s$ and $\boldsymbol{q} = \boldsymbol{q}_1^{(u)} \in \mathcal{F}^u$ with $u < s$,

$$\left\langle \Phi_{\boldsymbol{k}}, \Phi_{\boldsymbol{q}_1^{(u)}} \right\rangle = 0 \quad \forall \, u < s. \tag{S47}$$

Thus the $\Phi_{\boldsymbol{k}}$'s with $\boldsymbol{k} \in \mathcal{F}^s$ are orthonormal between each other and orthogonal to all $\Phi_{\boldsymbol{q}_1^{(u)}}$'s with $u < s$. Similarly, by taking $\boldsymbol{k} \in \mathcal{F}^u$ with $u < s$ and $\boldsymbol{q} \in \mathcal{F}^v$ with $v \leq u$, orthonormality is proven down to $\Phi_{\boldsymbol{k}_1^{(1)}}$. The zero-th eigenfunction $\Phi_{\boldsymbol{0}}(\boldsymbol{x}) = 1$ is also orthogonal to all other eigenfunctions by Eq. (S45) with $\boldsymbol{k} = 0$ and trivially normalised to 1.

Secondly, we prove that eigenfunctions from Eq. (S43) and eigenvalues from Eq. (S44) satisfy the kernel eigenproblem of $K^{CN}$. For $\boldsymbol{k} = \boldsymbol{0}$,

$$\int_{[0,1]^d} d^d y \, K^{CN}(\boldsymbol{x}, \boldsymbol{y}) = \frac{1}{d^2} \sum_{i,j=1}^d \int_{[0,1]^d} d^d y \sum_{\boldsymbol{q}} \lambda_{\boldsymbol{k}} e^{i\boldsymbol{q} \cdot (\boldsymbol{x}_i - \boldsymbol{y}_j)} = \lambda_{\boldsymbol{0}}, \qquad (S48)$$

proving that $\Lambda_{\boldsymbol{0}}$ and $\Phi_{\boldsymbol{0}}$ satisfy the eigenproblem. For $\boldsymbol{k} \neq \boldsymbol{0}$,

$$\int_{[0,1]^d} d^d y \, K^{CN}(\boldsymbol{x}, \boldsymbol{y}) \left( \frac{1}{\sqrt{d}} \sum_{l=1}^d e^{i\boldsymbol{k} \cdot \boldsymbol{y}_l} \right) = \frac{1}{d^{5/2}} \sum_{i,j,l=1}^d \int_{[0,1]^d} d^d y \sum_{\boldsymbol{q}} \lambda_{\boldsymbol{q}} e^{i\boldsymbol{q} \cdot (\boldsymbol{x}_i - \boldsymbol{y}_j)} e^{i\boldsymbol{k} \cdot \boldsymbol{y}_l}$$

$$= \frac{1}{d^{5/2}} \sum_{i=1}^d \sum_{\boldsymbol{q}} \lambda_{\boldsymbol{q}} e^{i\boldsymbol{q} \cdot \boldsymbol{x}_i} \sum_{j=1}^d \left( \delta(\boldsymbol{k}, \boldsymbol{q}) + \sum_{l \in \mathcal{P}_{j,+}} \delta(\boldsymbol{q}_1^{(s-o)}, \boldsymbol{0}) \, \delta(\boldsymbol{q}_{s-o+1}^{(o)}, \boldsymbol{k}_1^{(o)}) \, \delta(\boldsymbol{k}_{o+1}^{(s-o)}, \boldsymbol{0}) \right.$$

$$\left. + \sum_{l \in \mathcal{P}_{j,-}} \delta(\boldsymbol{k}_1^{(s-o)}, \boldsymbol{0}) \, \delta(\boldsymbol{q}_1^{(o)}, \boldsymbol{k}_{s-o+1}^{(o)}) \, \delta(\boldsymbol{q}_{o+1}^{(s-o)}, \boldsymbol{0}) \right).$$

$$(S49)$$

When $\boldsymbol{k} \in \mathcal{F}^s$, the deltas coming from the terms with $j \in \mathcal{P}_{j,\pm}$ vanish, showing that the eigenproblem is satisfied with $\Lambda_{\boldsymbol{k}} = \lambda_{\boldsymbol{k}}/d$ and $\Phi_{\boldsymbol{k}}(\boldsymbol{x}) = \sum_l e^{i\boldsymbol{k} \cdot \boldsymbol{x}}/\sqrt{d}$. When $\boldsymbol{k} \in \mathcal{F}^u$ with $u < s$, as the last $s - u$ components of $\boldsymbol{k}$ vanish, there are several $\boldsymbol{q}$'s satisfying the deltas in the bracket. There is $\boldsymbol{q} = \boldsymbol{k}$, from the $l = j$ term, then there are the $s - u$ $\boldsymbol{q}$'s such that $\delta(\boldsymbol{q}_1^{(s-o)}, \boldsymbol{0}) \delta(\boldsymbol{q}_{s-o+1}^{(o)}, \boldsymbol{k}_1^{(o)}) \delta(\boldsymbol{k}_{o+1}^{(s-o)}, \boldsymbol{0}) = 1$. These are all the $\boldsymbol{q}$'s having a $u$-dimensional patch equal to $\boldsymbol{k}_1^{(u)}$ and all the other elements set to zero, thus there are $(s - u + 1)$ such $\boldsymbol{q}$'s. Moreover, as $\lambda_{\boldsymbol{q}}$ depends only on the modulus of $\boldsymbol{q}$, all these $\boldsymbol{q}$'s result in the same eigenvalue, and in the same eigenfunction $\sum_l e^{i\boldsymbol{q} \cdot \boldsymbol{x}}/\sqrt{d}$, after the sum over patches. Therefore,

$$\int_{[0,1]^d} d^d y \, K^{CN}(\boldsymbol{x}, \boldsymbol{y}) \Phi_{\boldsymbol{k}_1^{(u)}} = \frac{(s - u + 1)}{d} \lambda_{\boldsymbol{k}_1^{(u)}} \Phi_{\boldsymbol{k}_1^{(u)}} = \Lambda_{\boldsymbol{k}_1^{(u)}} \Phi_{\boldsymbol{k}_1^{(u)}}. \qquad (S50)$$

Finally, we prove the expansion of the kernel in Eq. (S42),

$$K^{CN}(\boldsymbol{x}, \boldsymbol{y}) = \frac{1}{d^2} \sum_{i,j \in \mathcal{P}} C(\boldsymbol{x}_i, \boldsymbol{y}_j) \qquad (S51)$$

$$= \sum_{\boldsymbol{k}} \frac{1}{d} \lambda_{\boldsymbol{k}} \left( \frac{1}{\sqrt{d}} \sum_{i \in \mathcal{P}} \phi_{\boldsymbol{k}}(\boldsymbol{x}_i) \right) \overline{\left( \frac{1}{\sqrt{d}} \sum_{j \in \mathcal{P}} \phi_{\boldsymbol{k}}(\boldsymbol{y}_j) \right)}. \qquad (S52)$$

The terms on the RHS of Eq. (S51) are trivially equal to those of Eq. (S42) for $\boldsymbol{k} \in \mathcal{F}^s$. All the $\boldsymbol{k}$ having $s - u$ vanishing extremal components can be written as shifts of $\boldsymbol{k}_1^{(u)} \in \mathcal{F}^u$, which has the *last* $s - u$ components vanishing. But a shift of $\boldsymbol{k}$ does not affect $\lambda_{\boldsymbol{k}}$ nor $\sum_l e^{i\boldsymbol{k} \cdot \boldsymbol{x}}$, leading to a degeneracy of eigenvalues having $\boldsymbol{k}$ which can be obtained from a shift of $\boldsymbol{k}_1^{(u)} \in \mathcal{F}^u$. Such degeneracy is removed by restricting the sum over $\boldsymbol{k}$ to the sets $\mathcal{F}^u$, $u \leq s$, of wavevectors with non-vanishing extremal components, and rescaling the remaining eigenvalues with a factor of $(s - u + 1)/d$, so that Eq. (S42) is obtained. ∎

**Lemma C.2** (Spectra of overlapping local kernels). *Let $K^{LC}$ be a local kernel defined as in Eq. (14b), with $\mathcal{P} = \{1, \ldots, d\}$ and constituent kernel $C$ satisfying assumptions i), ii) above. Then, $K^{LC}$ admits the following Mercer's decomposition,*

$$K^{LC}(\boldsymbol{x}, \boldsymbol{y}) = \Lambda_{\boldsymbol{0}} + \sum_{u=1}^{s} \left( \sum_{\boldsymbol{k} \in \mathcal{F}^u} \sum_{i=1}^{d} \Lambda_{\boldsymbol{k},i} \Phi_{\boldsymbol{k},i}(\boldsymbol{x}) \Phi_{\boldsymbol{k},i}(\boldsymbol{y}) \right) \tag{S53}$$

*with eigenfunctions*

$$\Phi_{\boldsymbol{0}}(\boldsymbol{x}) = 1, \quad \Phi_{\boldsymbol{k},i}(\boldsymbol{x}) = \phi_{\boldsymbol{k}}(\boldsymbol{x}_i) \quad \forall \boldsymbol{k} \in \mathcal{F}^u \text{ with } 1 \leq u \leq s \text{ and } i = 1, \ldots, d, \tag{S54}$$

*and eigenvalues*

$$\Lambda_{\boldsymbol{0}} = \lambda_{\boldsymbol{0}}, \Lambda_{\boldsymbol{k},i} = \frac{s - u + 1}{d} \lambda_{\boldsymbol{k}} \quad \forall \boldsymbol{k} \in \mathcal{F}^u \text{ with } u \leq s \text{ and } i = 1, \ldots, d. \tag{S55}$$

*Proof.* We start by proving the orthonormality of the eigenfunctions. The scalar product $\langle \Phi_{\boldsymbol{k},i}, \Phi_{\boldsymbol{q},j} \rangle$ depends on the relation between the $i$-th and $j$-th patches.

$$\int_{[0,1]^d} d^d x \, \phi_{\boldsymbol{k}}(\boldsymbol{x}_i) \overline{\phi_{\boldsymbol{q}}(\boldsymbol{x}_j)}$$

$$= \delta(\boldsymbol{k}_1^{(s-o)}, \boldsymbol{0}) \, \delta(\boldsymbol{k}_{s-o+1}^{(o)}, \boldsymbol{q}_1^{(o)}) \, \delta(\boldsymbol{q}_{o+1}^{(s-o)}, \boldsymbol{0}), \quad \text{if } j \in \mathcal{P}_{i,+}, \tag{S56a}$$

$$= \delta(\boldsymbol{q}_1^{(s-o)}, \boldsymbol{0}) \, \delta(\boldsymbol{k}_1^{(o)}, \boldsymbol{q}_{s-o+1}^{(o)}) \, \delta(\boldsymbol{k}_{o+1}^{(s-o)}, \boldsymbol{0}), \quad \text{if } j \in \mathcal{P}_{i,-}, \tag{S56b}$$

$$= \delta(\boldsymbol{k}, \boldsymbol{0}) \, \delta(\boldsymbol{q}, \boldsymbol{0}), \qquad\qquad\qquad\qquad \text{if } j \notin \mathcal{P}_i, \tag{S56c}$$

$$= \delta(\boldsymbol{k}, \boldsymbol{q}), \qquad\qquad\qquad\qquad\qquad\quad \text{if } j = i. \tag{S56d}$$

Clearly, $\langle \Phi_{\boldsymbol{0}}, \Phi_{\boldsymbol{0}} \rangle = 1$ and setting one of $\boldsymbol{q}$ and $\boldsymbol{k}$ to $\boldsymbol{0}$ in Eq. (S56) yields orthogonality between $\Phi_{\boldsymbol{0}}$ and $\Phi_{\boldsymbol{k},i}$ for all $\boldsymbol{k} \neq \boldsymbol{0}$ and $i = 1, \ldots, d$. For any $\boldsymbol{k}$ and $\boldsymbol{q} \neq \boldsymbol{0}$, Eq. (S56d) implies

$$\langle \Phi_{\boldsymbol{k},i}, \Phi_{\boldsymbol{q},j} \rangle = \delta(\boldsymbol{k}, \boldsymbol{q}) \delta_{i,j} \tag{S57}$$

unless $\boldsymbol{k} = \boldsymbol{k}_1^{(u)} \in \mathcal{F}^u$ and $\boldsymbol{q}$ is a shift of $\boldsymbol{k}^{(u)}$. But such a $\boldsymbol{q}$ would have $q_1 = 0$ and there is no eigenfunction $\Phi_{\boldsymbol{q}}$ with $q_1 = 0$, apart from $\Phi_{\boldsymbol{0}}$. Hence, orthonormality is proven.

We then prove that eigenfunctions and eigenvalues defined in Eq. (S54) and Eq. (S55) satisfy the kernel eigenproblem of $K^{LC}$. For $\boldsymbol{k} = \boldsymbol{0}$,

$$\int_{[0,1]^d} d^d y \, K^{LC}(\boldsymbol{x}, \boldsymbol{y}) = \frac{1}{d} \sum_{i=1}^{d} \int_{[0,1]^d} d^d y \sum_{\boldsymbol{q}} \lambda_{\boldsymbol{k}} e^{i \boldsymbol{q} \cdot (\boldsymbol{x}_i - \boldsymbol{y}_i)} = \lambda_{\boldsymbol{0}}. \tag{S58}$$

In general,

$$\int_{[0,1]^d} d^d y \, K^{LC}(\boldsymbol{x}, \boldsymbol{y}) e^{i \boldsymbol{k} \cdot \boldsymbol{y}_l} = \frac{1}{d} \sum_{i=1}^{d} \int_{[0,1]^d} d^d y \sum_{\boldsymbol{q}} \lambda_{\boldsymbol{q}} e^{i \boldsymbol{q} \cdot (\boldsymbol{x}_i - \boldsymbol{y}_i)} e^{i \boldsymbol{k} \cdot \boldsymbol{y}_l}$$

$$= \frac{1}{d} \sum_{\boldsymbol{q}} \lambda_{\boldsymbol{q}} \Bigg( \delta(\boldsymbol{k}, \boldsymbol{q}) e^{i \boldsymbol{k} \cdot \boldsymbol{x}_l} + \sum_{i \notin \mathcal{P}_l} \delta(\boldsymbol{q}, \boldsymbol{0}) \, \delta(\boldsymbol{k}, \boldsymbol{0})$$

$$+ \sum_{i \in \mathcal{P}_{l,+}} e^{i \boldsymbol{q} \cdot \boldsymbol{x}_i} \delta(\boldsymbol{k}_1^{(s-o)}, \boldsymbol{0}) \, \delta(\boldsymbol{k}_{s-o+1}^{(o)}, \boldsymbol{q}_1^{(o)}) \, \delta(\boldsymbol{q}_{o+1}^{(s-o)}, \boldsymbol{0})$$

$$+ \sum_{i \in \mathcal{P}_{l,-}} e^{i \boldsymbol{q} \cdot \boldsymbol{x}_i} \delta(\boldsymbol{q}_1^{(s-o)}, \boldsymbol{0}) \, \delta(\boldsymbol{k}_1^{(o)}, \boldsymbol{q}_{s-o+1}^{(o)}) \, \delta(\boldsymbol{k}_{o+1}^{(s-o)}, \boldsymbol{0}) \Bigg). \tag{S59}$$

For $\boldsymbol{k} \in \mathcal{F}^u$, with $u = 1, \ldots, s$, the deltas which set the first component of $\boldsymbol{k}$ to 0 are never satisfied, therefore

$$
\int_{[0,1]^d} d^d y\, K^{LC}(\boldsymbol{x}, \boldsymbol{y}) e^{i\boldsymbol{k} \cdot \boldsymbol{y}_l}
$$
$$
= \frac{1}{d} \sum_{\boldsymbol{q}} \lambda_{\boldsymbol{q}} \left( \delta(\boldsymbol{k}, \boldsymbol{q}) e^{i\boldsymbol{k} \cdot \boldsymbol{x}_i} + \sum_{i \in \mathcal{P}_{l,-}} e^{i\boldsymbol{q} \cdot \boldsymbol{x}_i} \delta(\boldsymbol{q}_1^{(s-o)}, \boldsymbol{0})\, \delta(\boldsymbol{k}_1^{(o)}, \boldsymbol{q}_{s-o+1}^{(o)})\, \delta(\boldsymbol{k}_{o+1}^{(s-o)}, \boldsymbol{0}) \right). \tag{S60}
$$

The second term in brackets vanishes for $\boldsymbol{k} \in \mathcal{F}^s$ and the eigenvalue equation is satisfied with $\lambda_{\boldsymbol{k},l} = \lambda_{\boldsymbol{k}}/d$. For $\boldsymbol{k} = \boldsymbol{k}_1^{(u)} \in \mathcal{F}^u$ with $u < s$, $\delta(\boldsymbol{k}_{o+1}^{(s-o)}, \boldsymbol{0}) = 1$ for any $o \geq u$. As a result of the remaining deltas, the RHS of Eq. (S60) becomes a sum over all $\boldsymbol{q}$'s which can be obtained from shifts of $\boldsymbol{k}_1^{(u)}$, which are $s - u + 1$ (including $\boldsymbol{k}_1^{(u)}$ itself). The patch $\boldsymbol{x}_i$ which is multiplied by $\boldsymbol{q}$ in the exponent is also a shift of $\boldsymbol{x}_l$, thus all the factors $e^{i\boldsymbol{q} \cdot \boldsymbol{x}_i}$ appearing in the sum coincide with $e^{i\boldsymbol{k}_1^{(u)} \cdot \boldsymbol{x}_i}$. As $\lambda_{\boldsymbol{q}}$ depends on the modulus of $\boldsymbol{q}$, all these terms correspond to the same eigenvalue, $\lambda_{\boldsymbol{k}_1^{(u)}}$, so that

$$
\int_{[0,1]^d} d^d y\, K^{LC}(\boldsymbol{x}, \boldsymbol{y}) e^{i\boldsymbol{k}_1^{(u)} \cdot \boldsymbol{y}_l} = \left( \frac{s - u + 1}{d} \lambda_{\boldsymbol{k}_1^{(u)}} \right) e^{i\boldsymbol{k}_1^{(u)} \cdot \boldsymbol{x}_l}. \tag{S61}
$$

Finally, we prove the expansion of the kernel in Eq. (S53),

$$
K^{LC}(\boldsymbol{x}, \boldsymbol{y}) = \frac{1}{d} \sum_{i \in \mathcal{P}} C(\boldsymbol{x}_i, \boldsymbol{y}_i) = \sum_{\boldsymbol{k}} \frac{1}{d} \lambda_{\boldsymbol{k}} \sum_{i \in \mathcal{P}} \phi_{\boldsymbol{k}}(\boldsymbol{x}_i) \overline{\phi_{\boldsymbol{k}}(\boldsymbol{y}_i)}. \tag{S62}
$$

As in the proof of the eigendecomposition of convolutional kernels, all the $\boldsymbol{k}$ having $s - u$ vanishing extremal components can be written as shifts of $\boldsymbol{k}_1^{(u)} \in \mathcal{F}^u$, which has the *last* $s - u$ components vanishing. The shift of $\boldsymbol{k}$ does not affect $\lambda_{\boldsymbol{k}}$ nor the product $\phi_{\boldsymbol{k}}(\boldsymbol{x}_i) \overline{\phi_{\boldsymbol{k}}(\boldsymbol{y}_i)}$, after summing over $i$ leading to a degeneracy of eigenvalues which is removed by restricting the sum over $\boldsymbol{k}$ to the sets $\mathcal{F}^u$, $u \leq s$, and rescaling the remaining eigenvalues $\lambda_{\boldsymbol{k}_1^{(u)}}$ with a factor of $(s - u + 1)/d$, leading to Eq. (S53). ■

## D  Proof of Theorem 4.1

**Theorem D.1** (Theorem 4.1 in the main text). *Let $K_T$ be a d-dimensional convolutional kernel with a translationally-invariant t-dimensional constituent and leading nonanalyticity at the origin controlled by the exponent $\alpha_t > 0$. Let $K_S$ be a d-dimensional convolutional or local student kernel with a translationally-invariant s-dimensional constituent, and with a nonanalyticity at the origin controlled by the exponent $\alpha_s > 0$. Assume, in addition, that if the kernels have overlapping patches then $s \geq t$; whereas if the kernels have nonoverlapping patches s is an integer multiple of t; and that data are uniformly distributed on a d-dimensional torus. Then, the following asymptotic equivalence holds in the limit $P \to \infty$,*

$$
\mathcal{B}(P) \sim P^{-\beta}, \quad \beta = \alpha_t/s. \tag{S63}
$$

*Proof.* For the sake of clarity, we start with the proof in the nonoverlapping-patches case, and then extend it to the overlapping-patches case. Since $K_T$ and $K_S$ have translationally-invariant constituent kernels and data are uniformly distributed on a $d$-dimensional torus, the kernels can be diagonalised in Fourier space. Let us start by considering a convolutional student: because of the constituent kernel's isotropy, the Fourier coefficients $\Lambda_{\boldsymbol{k}}^{(s)}$ of $K_S$ depend on $k$ (modulus of $\boldsymbol{k}$) only. Notice the superscript indicating the dimensionality of the student constituents. In particular, $\Lambda_{\boldsymbol{k}}^{(s)}$ is a decreasing function of $k$ and, for large $k$, $\Lambda_{\boldsymbol{k}} \sim k^{-(s+\alpha_s)}$. Then, $\mathcal{B}(P)$ reads

$$
\mathcal{B}(P) = \sum_{\{\boldsymbol{k} \,|\, k > k_c(P)\}} \mathbb{E}[|c_{\boldsymbol{k}}|^2], \tag{S64}
$$

where $k_c(P)$ is defined as the wavevector modulus of the $P$-th largest eigenvalue and $\mathbb{E}[|c_{\boldsymbol{k}}|^2]$ denotes the variance of the target coefficients in the student eigenbasis. $k_c(P)$ is such that there are exactly $P$ eigenvalues with $k \leq k_c(P)$,

$$P = \sum_{\{\boldsymbol{k}|k<k_c(P)\}} 1 \sim \int \frac{d^s k}{(2\pi)^s} \theta(k_c(P) - k) = \frac{1}{(2\pi)^s} \frac{\pi^{s/2}}{\Gamma(s/2+1)} k_c(P)^s, \qquad \text{(S65)}$$

i.e. $k_c(P) \sim P^{1/s}$.

By denoting the eigenfunctions of the student with $\Phi_{\boldsymbol{k}}^{(s)}$, the superscript $(s)$ indicating the dimension of the constituent's plane waves,

$$\mathbb{E}[|c_{\boldsymbol{k}}|^2] = \int_{[0,1]^d} d^d x \, \Phi_{\boldsymbol{k}}^{(s)}(\boldsymbol{x}) \int_{[0,1]^d} d^d y \, \overline{\Phi_{\boldsymbol{k}}^{(s)}(\boldsymbol{y})} \mathbb{E}[f^*(\boldsymbol{x}) f^*(\boldsymbol{y})] \qquad \text{(S66)}$$

$$= \int_{[0,1]^d} d^d x \, \Phi_{\boldsymbol{k}}^{(s)}(\boldsymbol{x}) \int_{[0,1]^d} d^d y \, \overline{\Phi_{\boldsymbol{k}}^{(s)}(\boldsymbol{y})} K_T(\boldsymbol{x}, \boldsymbol{y}).$$

Decomposing the teacher kernel $K_T$ into its eigenvalues $\Lambda_{\boldsymbol{q}}^{(t)}$ and eigenfunctions $\Phi_{\boldsymbol{q}}^{(t)}(\boldsymbol{y})$,

$$\mathbb{E}[|c_{\boldsymbol{k}}|^2] = \int_{[0,1]^d} d^d x \, \Phi_{\boldsymbol{k}}^{(s)}(\boldsymbol{x}) \int_{[0,1]^d} d^d y \, \overline{\Phi_{\boldsymbol{k}}^{(s)}(\boldsymbol{y})} \Bigg( \Lambda_{\boldsymbol{0}}^{(t)} \qquad \text{(S67)}$$

$$+ \frac{s}{d} \sum_{\boldsymbol{q} \neq \boldsymbol{0}} \Lambda_{\boldsymbol{q}}^{(t)} \sum_{i \in \mathcal{P}^{(t)}} \phi_{\boldsymbol{q}}^{(t)}(\boldsymbol{x}_i) \sum_{j \in \mathcal{P}^{(t)}} \overline{\phi_{\boldsymbol{q}}^{(t)}(\boldsymbol{y}_j)} \Bigg).$$

The $\boldsymbol{q} = \boldsymbol{0}$ mode of the teacher can give non-vanishing contributions to $c_{\boldsymbol{0}}$ only, therefore it does not enter any term of the sum in Eq. (S64). Once we removed the term with $\boldsymbol{q} = \boldsymbol{0}$, consider the $\boldsymbol{y}$-integral:

$$\mathcal{I}_{\boldsymbol{k}}(\boldsymbol{x}) = \int_{[0,1]^d} d^d y \sqrt{\frac{s}{d}} \sum_{m \in \mathcal{P}^{(s)}} \overline{\phi_{\boldsymbol{k}}^{(s)}(\boldsymbol{y}_m)} \frac{s}{d} \sum_{\boldsymbol{q} \neq \boldsymbol{0}} \Lambda_{\boldsymbol{q}}^{(t)} \sum_{i \in \mathcal{P}^{(t)}} \phi_{\boldsymbol{q}}^{(t)}(\boldsymbol{x}_i) \sum_{j \in \mathcal{P}^{(t)}} \overline{\phi_{\boldsymbol{q}}^{(t)}(\boldsymbol{y}_j)} \qquad \text{(S68)}$$

$$= \left(\frac{s}{d}\right)^{\frac{3}{2}} \sum_{\boldsymbol{q} \neq \boldsymbol{0}} \Lambda_{\boldsymbol{q}}^{(t)} \sum_{i \in \mathcal{P}^{(t)}} \phi_{\boldsymbol{q}}^{(t)}(\boldsymbol{x}_i) \sum_{m \in \mathcal{P}^{(s)}} \sum_{j \in \mathcal{P}^{(t)}} \int_{[0,1]^d} d^d y \, \overline{\phi_{\boldsymbol{k}}^{(s)}(\boldsymbol{y}_m)} \, \overline{\phi_{\boldsymbol{q}}^{(t)}(\boldsymbol{y}_j)}.$$

As all the $t$-dimensional patches of the teacher must be contained in at least one of the $s$-dimensional patches of the student, in the nonoverlapping case we require that $s$ is an integer multiple of $t$. Then, each of the teacher patches is entirely contained in one and only one patch of the student. If the teacher patch $\boldsymbol{y}_j$ is not contained in the student patch $\boldsymbol{y}_m$, we can factorise the integration over $\boldsymbol{y}$ into two integrals over $\boldsymbol{y}_j$ and $\boldsymbol{y}_m$. These terms give vanishing contributions to $\mathcal{I}_{\boldsymbol{k}}(\boldsymbol{x})$ since the integral of a plane wave over a period is always zero for non-zero wavevectors. Instead, if the teacher patch $\boldsymbol{y}_j$ is contained in the student patch $\boldsymbol{y}_m$, denoting with $l$ the index of the element of $\boldsymbol{y}_m$ which coincide with the first element of $\boldsymbol{y}_j$, we can factorise the student eigenfunctions as follows

$$\phi_{\boldsymbol{k}}^{(s)}(\boldsymbol{y}_m) = \phi_{\boldsymbol{k}_l^{(t)}}^{(t)}(\boldsymbol{y}_j) \phi_{\boldsymbol{k} \smallsetminus \boldsymbol{k}_l^{(t)}}^{(s-t)}(\boldsymbol{y}_{m \smallsetminus j}). \qquad \text{(S69)}$$

Here $\boldsymbol{k}_l^{(t)}$ denotes the $t$-dimensional patch of $\boldsymbol{k}$ starting at $l$ and $\boldsymbol{k} \smallsetminus \boldsymbol{k}_l^{(t)}$ the sequence of elements which are in $\boldsymbol{k}$ but not in $\boldsymbol{k}_l^{(t)}$. As $s$ is an integer multiple of $t$, $l = \tilde{l} \times s/t$ with $\tilde{l} = 1, \ldots, t$. Inserting Eq. (S69) into Eq. (S68),

$$\mathcal{I}_{\boldsymbol{k}}(\boldsymbol{x}) = \sum_{l = \tilde{l} s/t, \, \tilde{l} = 1}^{t} \delta(\boldsymbol{k} \smallsetminus \boldsymbol{k}_l^{(t)}, \boldsymbol{0}) \Lambda_{\boldsymbol{k}_l^{(t)}}^{(t)} \sqrt{\frac{s}{d}} \sum_{i \in \mathcal{P}^{(t)}} \overline{\phi_{\boldsymbol{k}_l^{(t)}}^{(t)}(\boldsymbol{x}_i)}. \qquad \text{(S70)}$$

The $x$-integral of Eq. (S66) can be performed by the same means after expanding $\Phi_{\boldsymbol{k}}^{(s)}$ as a sum of $s$-dimensional plane waves, so as to get,

$$\mathbb{E}[|c_{\boldsymbol{k}}|^2] = \sum_{l=\tilde{l}s/t, \tilde{l}=1}^{t} \delta(\boldsymbol{k} \smallsetminus \boldsymbol{k}_l^{(t)}, \boldsymbol{0}) \, \Lambda_{\boldsymbol{k}_l^{(t)}}^{(t)}. \tag{S71}$$

Therefore, $\mathbb{E}[|c_{\boldsymbol{k}}|^2]$ is non-zero only for $\boldsymbol{k}$'s which have at most $t$ consecutive components greater or equal than zero, and the remaining $s - t$ being strictly zero. Inserting Eq. (S71) into Eq. (S64),

$$\mathcal{B}(P) = \sum_{\{\boldsymbol{k}|k>k_c(P)\}} \sum_{l=\tilde{l}s/t, \tilde{l}=1}^{t} \delta(\boldsymbol{k} \smallsetminus \boldsymbol{k}_l^{(t)}, \boldsymbol{0}) \, \Lambda_{\boldsymbol{k}_l^{(t)}}^{(t)} \sim \int_{P^{1/s}}^{\infty} dk \, k^{t-1} k^{-(\alpha_t+t)} \sim P^{-\frac{\alpha_t}{s}}. \tag{S72}$$

When using a local student, the convolutional eigenfunctions in the RHS of Eq. (S66) are replaced by the local eigenfunctions $\Phi_{\boldsymbol{k},i}(\boldsymbol{x})$ of Eq. (18). Repeating the same computations, one finds

$$k_c \sim \left( \frac{P}{d/s} \right)^{\frac{1}{s}}, \tag{S73}$$

$$\mathbb{E}[|c_{\boldsymbol{k},i}|^2] = \frac{s}{d} \sum_{l=\tilde{l}s/t, \tilde{l}=1}^{t} \delta(\boldsymbol{k} \smallsetminus \boldsymbol{k}_l^{(t)}, \boldsymbol{0}) \, \Lambda_{\boldsymbol{k}_l^{(t)}}^{(t)}. \tag{S74}$$

As a result,

$$\mathcal{B}(P) = \sum_{i \in \mathcal{P}} \sum_{\{\boldsymbol{k}|k>k_c(P)\}} \frac{s}{d} \sum_{l=\tilde{l}s/t, \tilde{l}=1}^{t} \delta(\boldsymbol{k} \smallsetminus \boldsymbol{k}_l^{(t)}, \boldsymbol{0}) \, \Lambda_{\boldsymbol{k}_l^{(t)}}^{(t)} \tag{S75}$$

$$\sim \int_{\left( \frac{P}{d/s} \right)^{\frac{1}{s}}}^{\infty} dk \, k^{t-1} k^{-(\alpha_t+t)} \sim \left( \frac{P}{d/s} \right)^{-\frac{\alpha_t}{s}}. \tag{S76}$$

As we showed in Appendix C, when the patches overlap the set of wavevectors which index the eigenvalues is restricted from $\mathbb{Z}^s$ to the union of the $\mathcal{F}^u$'s for $u = 0, \ldots, s$. In addition, the eigenvalues with $\boldsymbol{k} \in \mathcal{F}^u$, $0 < u < s$, are rescaled by a factor $(s - u + 1)/d$. Therefore, in the overlapping case the eigenvalues do not decrease monotonically with $k$ and $\mathcal{B}(P)$ cannot be written as a sum of over $\boldsymbol{k}$'s with modulus $k$ larger than a certain threshold $k_c$. By considering also that, with $t \leq s$, $\mathbb{E}[|c_{\boldsymbol{k}}|^2]$ is non-zero only for $\boldsymbol{k}$'s which have at most $t$ consecutive nonvanishing components, we have

$$\mathcal{B}(P) = \sum_{u=0}^{t} \sum_{\boldsymbol{k} \in \mathcal{F}^u} \mathbb{E}[|c_{\boldsymbol{k}}|^2] \chi(\Lambda_{\boldsymbol{k}}^{(s)} > \Lambda_P), \tag{S77}$$

where $\Lambda_P$ denotes the $P$-th largest eigenvalue and the indicator function $\chi(\Lambda_{\boldsymbol{k}}^{(s)} > \Lambda_P)$ ensures that the sum runs over all but the first $P$ eigenvalues of the student. The sets $\{\mathcal{F}^u\}_{u<t}$ have all null measure in $\mathbb{Z}^t$, whereas $\mathcal{F}^t$ is dense in $\mathbb{Z}^t$, thus the asymptotics of $\mathcal{B}(P)$ are dictated by the sum over $\mathcal{F}^t$. When $\boldsymbol{k}$'s are restricted to the latter set, eigenvalues are again decreasing functions of $k$ and the constraint $\Lambda_{\boldsymbol{k}}^{(s)} > \Lambda_P$ translates into $k > k_c(P)$. Having changed, with respect to the nonoverlapping case, only an infinitesimal fraction of the eigenvalues, the asymptotic scaling of $k_c(P)$ with $P$ remains unaltered and the estimates of Eq. (S72) and Eq. (S74) extend to kernels with nonoverlapping patches after substituting the degeneracy $d/s$ with $|\mathcal{P}| = d$.

∎

# E    Asymptotic learning curves with a local teacher

**Theorem E.1.** *Let $K_T$ be a $d$-dimensional local kernel with a translationally-invariant $t$-dimensional constituent and leading nonanalyticity at the origin controlled by the exponent $\alpha_t > 0$. Let $K_S$ be a*

*d-dimensional local student kernel with a translationally-invariant s-dimensional constituent, and with a nonanalyticity at the origin controlled by the exponent $\alpha_s > 0$. Assume, in addition, that if the kernels have overlapping patches then $s \geq t$; whereas if the kernels have nonoverlapping patches $s$ is an integer multiple of $t$; and that data are uniformly distributed on a $d$-dimensional torus. Then, the following asymptotic equivalence holds in the limit $P \to \infty$,*

$$\mathcal{B}(P) \sim P^{-\beta}, \quad \beta = \alpha_t/s. \tag{S78}$$

*Proof.* The proof is analogous to that of Appendix D, the only difference being that eigenfunctions and eigenvalues are indexed by $\boldsymbol{k}$ and the patch index $i$. This results in an additional factor of $d/s$ in the RHS of Eq. (S65). All the discussion between Eq. (S66) and Eq. (S71) can be repeated by attaching the additional patch index $i$ to all coefficients. Eq. (S72) for $\mathcal{B}(P)$ is then corrected with an additional sum over patches. The extra sum, however, does not influence the asymptotic scaling with $P$. ∎

## F    Proof of Theorem 6.1

**Theorem F.1** (Theorem 6.1 in the main text). *Let us consider a positive-definite kernel $K$ with eigenvalues $\Lambda_\rho$, $\sum_\rho \Lambda_\rho < \infty$, and eigenfunctions $\Phi_\rho$ learning a (random) target function $f^*$ in kernel ridge regression (Eq. (3)) with ridge $\lambda$ from $P$ observations $f^*_\mu = f^*(\boldsymbol{x}^\mu)$, with $\boldsymbol{x}^\mu \in \mathbb{R}^d$ drawn from a certain probability distribution. Let us denote with $\mathcal{D}_T(\Lambda)$ the reduced density of kernel eigenvalues with respect to the target and $\epsilon(\lambda, P)$ the generalisation error and also assume that*

*i) For any $P$-tuple of indices $\rho_1, \ldots, \rho_P$, the vector $(\Phi_{\rho_1}(\boldsymbol{x}^1), \ldots, \Phi_{\rho_P}(\boldsymbol{x}^P))$ is a Gaussian random vector;*

*ii) The target function can be written in the kernel eigenbasis with coefficients $c_\rho$ and $c^2(\Lambda_\rho) = \mathbb{E}[|c_\rho|^2]$, with $\mathcal{D}_T(\Lambda) \sim \Lambda^{-(1+r)}$, $c^2(\Lambda) \sim \Lambda^q$ asymptotically for small $\Lambda$ and $r > 0$, $r < q < r + 2$;*

*Then the following equivalence holds in the joint $P \to \infty$ and $\lambda \to 0$ limit with $1/(\lambda\sqrt{P}) \to 0$:*

$$\epsilon(\lambda, P) \sim \sum_{\{\rho | \Lambda_\rho < \lambda\}} \mathbb{E}[|c_\rho|^2] = \int_0^\lambda d\Lambda \mathcal{D}_T(\Lambda) c^2(\Lambda). \tag{S79}$$

*Proof.* In this proof we make use of results derived in [21]. Our setup for kernel ridge regression correspond to what the authors of [21] call the *classical setting*. Let us introduce the integral operator $T_K$ associated with the kernel, defined by

$$(T_K f)(\boldsymbol{x}) = \int p\left(d^d y\right) K(\boldsymbol{x}, \boldsymbol{y}) f(\boldsymbol{y}). \tag{S80}$$

The trace $Tr[T_K]$ coincide with the sum of $K$'s eigenvalues and is finite by hypothesis. We define the following estimator of the generalisation error $\epsilon(\lambda, P)$,

$$\mathcal{R}(\lambda, P) = \partial_\lambda \vartheta(\lambda) \int p(d^d x) \left(f^*(\boldsymbol{x}) - (\mathcal{A}_\vartheta f^*)(\boldsymbol{x})\right)^2, \tag{S81}$$

where $\vartheta(\lambda)$ is the *signal capture threshold* (SCT) [21] and $\mathcal{A}_\vartheta = T_K(T_K + \vartheta(\lambda))^{-1}$ is a reconstruction operator [21]. The target function can be written in the kernel eigenbasis by hypothesis (with coefficients $c_\rho$) and $T_K$ has the same eigenvalues and eigenfunctions of the kernel by definition. Hence,

$$\mathcal{R}(\lambda, P) = \partial_\lambda \vartheta(\lambda) \sum_{\rho=1}^\infty \frac{\vartheta(\lambda)^2}{(\Lambda_\rho + \vartheta(\lambda))^2} |c_\rho|^2 = \partial_\lambda \vartheta(\lambda) \int_0^\infty d\Lambda \, \mathcal{D}_T(\Lambda) c^2(\Lambda) \frac{\vartheta(\lambda)^2}{(\Lambda + \vartheta(\lambda))^2}, \tag{S82}$$

where $\mathcal{D}_T$ is the reduced density of eigenvalues Eq. (25). We now derive the asymptotics of $\mathcal{R}(\lambda, P)$ in the joint $P \to \infty$ and $\lambda \to 0$ limit, then relate the asymptotics of $\mathcal{R}$ to those of $\epsilon(\lambda, P)$ via a theorem proven in [21].

Proposition 3 of [21] shows that for any $\lambda > 0$, $\partial_\lambda \vartheta(\lambda) \to 1$ and $\vartheta(\lambda) \to \lambda$ with corrections of order $1/N$. Thus, we focus on the following integral,

$$\int_0^\infty d\Lambda \, \mathcal{D}_T(\Lambda) c^2(\Lambda) \frac{\lambda^2}{(\Lambda + \lambda)^2}. \tag{S83}$$

The functions $\mathcal{D}_T(\Lambda)$ and $c^2(\Lambda)$ can be safely replaced with their small-$\Lambda$ expansions $\Lambda^{-(1+r)}$ and $\Lambda^q$ over the whole range of the integral above because of the assumptions $q > r$ and $q \leq r + 2$. In practice, there should be an upper cut-off on the integral coinciding with the largest eigenvalue $\Lambda_1$, but the assumption $q \leq r + 2$ causes this part of the spectrum to be irrelevant for the asymptotics of the error. In fact, we will conclude that the integral is dominated by the portion of the domain around $\lambda$. After the change of variables $y = \Lambda/\lambda$,

$$\int_0^\infty d\Lambda \, \mathcal{D}_T(\Lambda) c^2(\Lambda) \frac{\lambda^2}{(\Lambda + \lambda)^2} = \lambda^{q-r} \int dy \, \frac{y^{q-1-r}}{(1+y)^2}, \tag{S84}$$

where one recognises one of the integral representations of the beta function,

$$B(a, b) = \int dy \, \frac{y^{a-1}}{(1+y)^{a+b}} = \frac{\Gamma(a)\Gamma(b)}{\Gamma(a+b)}, \tag{S85}$$

with $\Gamma$ denoting the gamma function. Therefore,

$$\int_0^\infty d\Lambda \, \mathcal{D}_T(\Lambda) c^2(\Lambda) \frac{\lambda^2}{(\Lambda + \lambda)^2} = \lambda^{q-r} \frac{\Gamma(q-r)\Gamma(2-q+r)}{\Gamma(2)}. \tag{S86}$$

It is interesting to notice how the assumptions $q > r$ and $q < r + 2$ are required in order to avoid the poles of the $\Gamma$ functions in the RHS of Eq. (S86).

We now use Eq. (S86) to infer the asymptotics of $\mathcal{R}(\lambda, P)$ in the scaling limit $\lambda \to 0$ and $P \to \infty$ with $1/(\lambda\sqrt{P}) \to 0$. The latter condition implies that $\lambda$ decays more slowly than $(P)^{-1/2}$, thus additional terms stemming from the finite-$P$ difference between $\vartheta$ and $\lambda$, of order $P^{-1}$ are negligible w.r.t. $\lambda^{q-r}$. The finite-$P$ difference between $\partial_\lambda \vartheta$, also $O(P^{-1})$, can be neglected too. Finally,

$$\mathcal{R}(\lambda, P) \sim \int_0^\infty d\Lambda \, \mathcal{D}_T(\Lambda) c^2(\Lambda) \frac{\lambda^2}{(\Lambda + \lambda)^2} \sim \lambda^{q-r} \sim \int_0^\lambda d\Lambda \mathcal{D}_T(\Lambda) c^2(\Lambda). \tag{S87}$$

Theorem 6 of [21] shows the convergence of $\epsilon(\lambda, P)$ towards $\mathcal{R}(\lambda, P)$ when $P \to \infty$. Specifically,

$$|\epsilon(\lambda, P) - \mathcal{R}(\lambda, P)| \leq \left( \frac{1}{P} + g\left( \frac{Tr[T_K]}{\lambda\sqrt{P}} \right) \right) \mathcal{R}(\lambda, P), \tag{S88}$$

where $g$ is a polynomial with non-negative coefficients and $g(0) = 0$. With a decaying ridge $\lambda(P)$ such that $1/(\lambda\sqrt{P}) \to 0$, both $\mathcal{R}/P$ and $\mathcal{R}g(Tr[T_K]/(\lambda\sqrt{P}))$ tend to zero faster than $\mathcal{R}$ itself, thus the asymptotics of $\epsilon(\lambda, P)$ coincide with those of $\mathcal{R}(\lambda, P)$ and Eq. (S79) is proven. ∎

**Remark** The estimate for the exponent $\beta$ of Corollary 6.1.1 follows from the theorem above with $r = t/(s + \alpha_s)$, $q = (\alpha_t + t)/(\alpha_s + s)$ and $\lambda \sim P^{-\gamma}$. Then $q > r$ because $\alpha_t > 0$, whereas the condition $q < r + 2$ is equivalent to the assumption $\alpha_t < 2(\alpha_s + s)$ required in Section 4 in order to derive the learning curve exponent in Eq. (20) from our estimate of $\mathcal{B}(P)$.

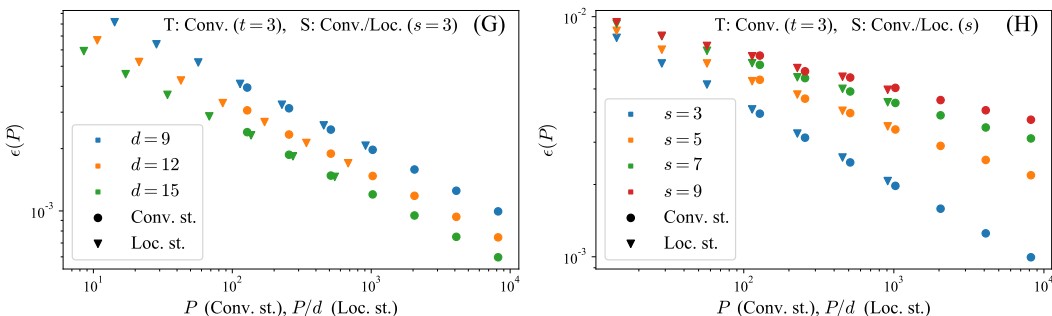

Figure S1: Learning curves for convolutional teacher and local and convolutional student kernels, with filter sizes denoted by $t$ and $s$ respectively. Data are sampled uniformly in the hypercube $[0, 1]^d$, with $d = 9$ if not specified otherwise. The sample complexity $P$ of the local students is rescaled with the number of patches to highlight the pre-asymptotic effect of shift-invariance on the learning curves.

## G    Numerical experiments

### G.1    Details on the simulations

To obtain the empirical learning curves, we generate $P + P_{\text{test}}$ random points uniformly distributed in a $d$-dimensional hypercube or on the surface of a $d - 1$-dimensional hypersphere embedded in $d$ dimensions. We use $P \in \{128, 256, 512, 1024, 2048, 4096, 8192\}$ and $P_{\text{test}} = 8192$. For each value of $P$, we generate a Gaussian random field with covariance given by the teacher kernel, and we compute the kernel ridgeless regression predictor of the student kernel using Eq. (4) with the $P$ training samples. The generalisation error defined in Eq. (5) is approximated by computing the empirical mean squared error on the $P_{\text{test}}$ unseen samples. The expectation with respect to the target function is obtained averaging over 128 independent teacher Gaussian processes, each sampled on different points of the domain. As teacher and student kernels, we consider different combinations of the convolutional and local kernels defined in Eq. (14a) and Eq. (14b), with Laplacian constituents $\mathcal{C}(\boldsymbol{x}_i - \boldsymbol{x}_j) = e^{-\|\boldsymbol{x}_i - \boldsymbol{x}_j\|}$ and overlapping patches. In particular,

- the cases with convolutional teacher and both convolutional and local students with various filter sizes are reported in Fig. 1 and Fig. S3 for data distributed in a hypercube and on a hypersphere respectively;

- the cases with local teacher and both local and convolutional students are reported in Fig. S2 for data distributed in a hypercube.

Experiments are run on NVIDIA Tesla V100 GPUs using the PyTorch package. The approximate total amount of time to reproduce all experiments with our setup is 400 hours. Code for reproducing the experiments can be found at `https://github.com/fran-cagnetta/local_kernels`.

### G.2    Additional experiments

**Convolutional vs local students**    In Fig. S1 we report the empirical learning curves for convolutional and local student kernels learning a convolutional teacher kernel, with filter sizes $s$ and $t$ respectively. Data are uniformly sampled in the hypercube $[0, 1]^d$. By rescaling the sample complexity $P$ of the local students with the number of patches $|\mathcal{P}| = d$, the learning curves of local and convolutional students overlap, confirming our prediction on the role of shift-invariance. Indeed, the local student has to learn the same local task at all the possible patch locations, while the convolutional student is naturally shift-invariant.

**Local teacher**    In Fig. S2 we report the empirical learning curves for a local teacher kernel and data uniformly sampled in the hypercube $[0, 1]^d$. In panels I and J, also the student is a local kernel and the same discussion of Section 5 applies. In panel K, the student is a convolutional kernel and the generalisation error does not decrease by increasing the size of the training set. Indeed, a local

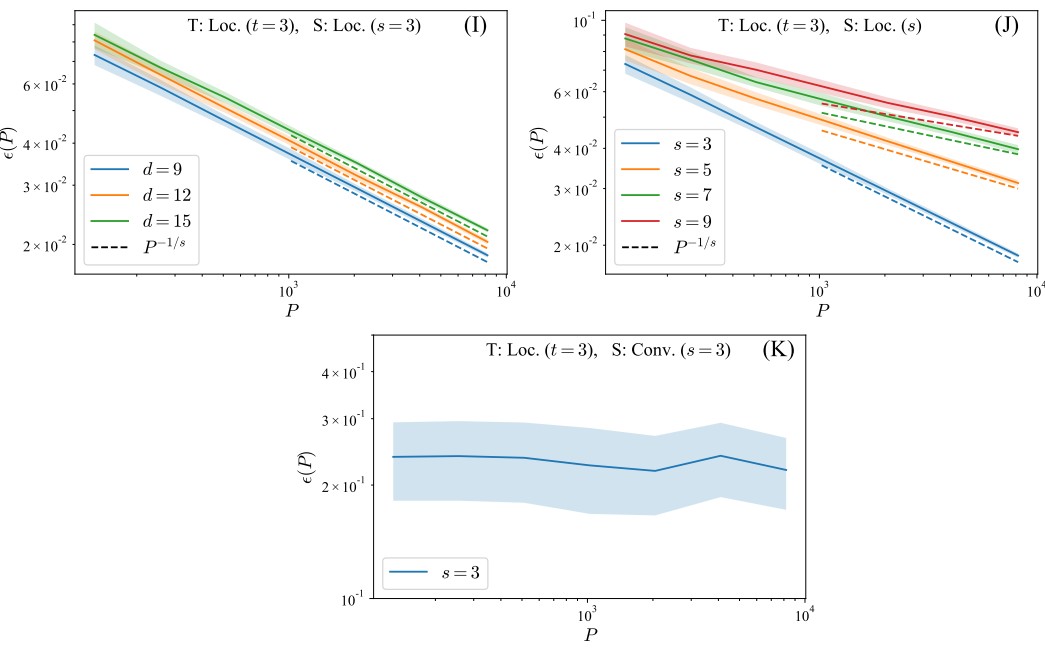

Figure S2: Learning curves for local teacher and local and convolutional student kernels, with filter sizes denoted by $t$ and $s$ respectively. Data are sampled uniformly in the hypercube $[0,1]^d$, with $d = 9$ if not specified otherwise. Solid lines are the results of numerical experiments averaged over 128 realisations and the shaded areas represent the empirical standard deviations. The predicted scaling are shown by dashed lines.

non-shift-invariant function is not on the span of the eigenfunctions of a convolutional kernel, and therefore the student is not able to learn the target.

**Spherical data**   In Fig. S3 we report the empirical learning curves for convolutional teacher and convolutional (left panels) and local (right panels) student kernels. Data are restricted to the unit sphere $\mathbb{S}^{d-1}$. Panels L-O are the analogous of panels A-D of Fig. 1. Notice that when the filter size of the student coincides with $d$ (panels P, Q), the learning curves decay with exponent $\beta = 1/(d-1)$ (instead of $\beta = 1/d$). Indeed, for data normalised on $\mathbb{S}^{d-1}$, the spectrum of the Laplacian kernel decays at a rate $\mathcal{O}(k^{-\alpha-(d-1)})$ with $\alpha = 1$. However, as the student filter size is lowered, we recover the exponent $1/s$ independently of the dimension $d$ of input space, as derived for data on the torus and shown empirically for data in the hypercube. In fact, we expect that the $s$-dimensional marginals of the uniform distribution on $\mathbb{S}^{d-1}$ become insensitive to the spherical constraint when $s \ll d$.

**Convolutional NTKs**   In Fig. S4 we report the empirical learning curves obtained using the NTK of one-hidden-layer CNNs with ReLU activations, which corresponds to using the kernel $\Theta^{FC}$ defined in Eq. (S14) as the constituent. Since this kernel is not translationally invariant, it cannot be diagonalised in the Fourier domain, and the analysis of Section 4 does not apply. However, as shown in panels P-S, the same learning curve exponents $\beta$ of the Laplacian-constituent case are recovered. Indeed, $\Theta^{FC}$ and the Laplacian kernel share the same nonanalytic behaviour in the origin, and their spectra have the same asymptotic decay [32]. In Fig. S5 we present the same plots of panels R and S, but instead of the analytical NTKs, we compute numerically the kernels of randomly-initialised very-wide CNNs ($H \approx 10^6$).

**Real data**   In Fig. Fig. S6 we report the learning curves of local kernels with Laplacian constituents applied to the CIFAR-10 dataset. We build the tasks by selecting two classes and assigning label $+1$ to data from one class and $-1$ to data from the other class. As before, we use $P \in \{128, 256, 512, 1024, 2048, 4096, 8192\}$ and $P_{\text{test}} = 8192$. Differently from our assumptions, image data are strongly anisotropic, and the distance between nearest-neighbour points decays faster than $P^{-1/d}$. Indeed, target functions defined on data of this kind are usually not cursed with the

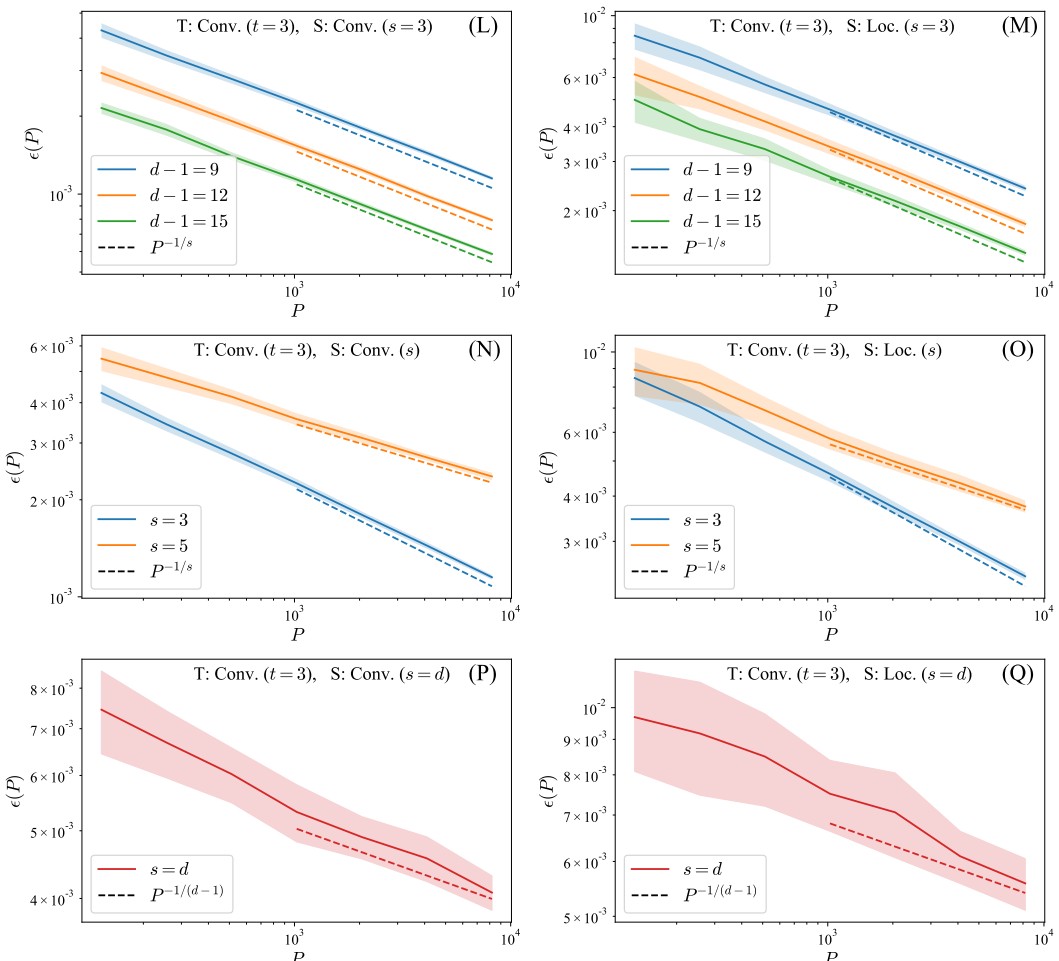

Figure S3: Learning curves for data uniformly distributed on the unit sphere $\mathbb{S}^{d-1}$, with $d = 10$ if not specified otherwise. The teacher and student filter sizes are denoted with $t$ and $s$ respectively. Solid lines are the results of numerical experiments averaged over 128 realisations and the shaded areas represent the empirical standard deviations.

full dimensionality $d$ of the inputs, but rather with an effective dimension $d_{\text{eff}}$. $d_{\text{eff}}$ is related to the dimension of the manifold in which data lie [4], and may also vary when extracting patches of different sizes. Nonetheless, as we found in our synthetic setup, the learning curve exponent $\beta$ increases monotonically with the filter size of the kernel, strengthening the concept that leveraging locality is key for performance.

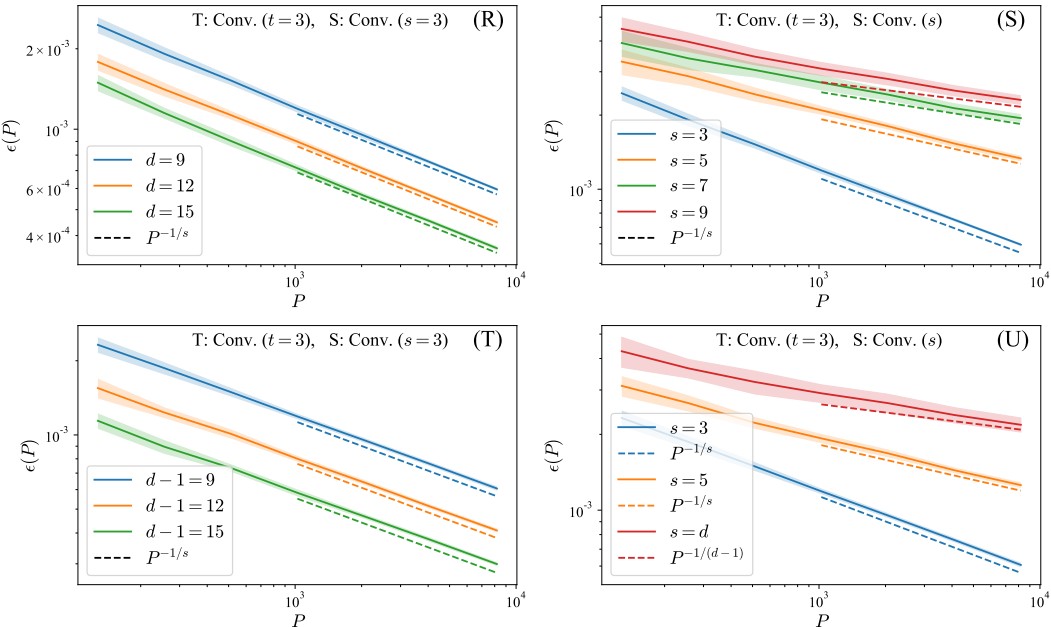

Figure S4: Learning curves for convolutional NTKs and data uniformly distributed in the hypercube $[0,1]^d$ (panels R, S) or on the unit sphere $\mathbb{S}^{d-1}$ (panels T, U). The teacher and student filter sizes are denoted with $t$ and $s$ respectively. Solid lines are the results of numerical experiments averaged over 128 realisations and the shaded areas represent the empirical standard deviations.

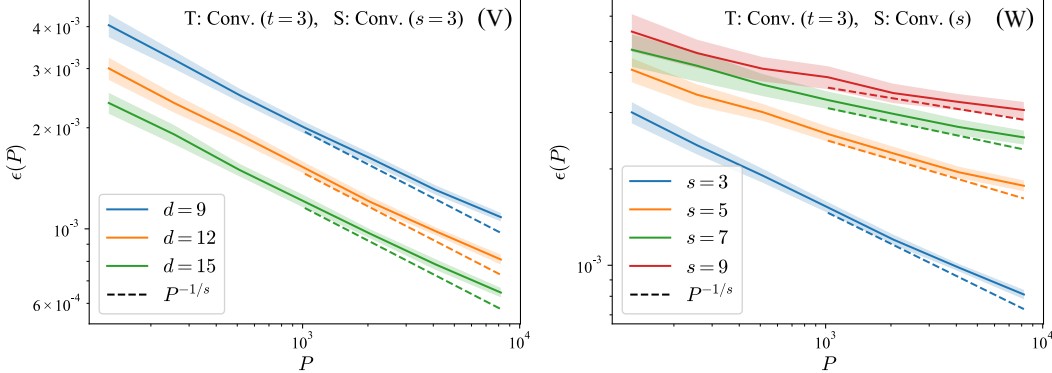

Figure S5: Learning curves for empirical NTKs of very-wide one-hidden-layer CNNs ($H \approx 10^6$) and data uniformly distributed in the hypercube $[0,1]^d$. The teacher and student filter sizes are denoted with $t$ and $s$ respectively. Solid lines are the results of numerical experiments averaged over 128 realisations and the shaded areas represent the empirical standard deviations.

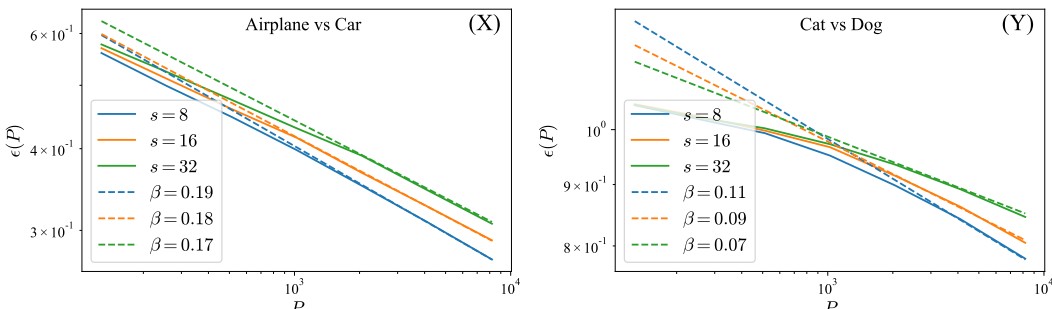

Figure S6: Learning curves of local kernels with filters of size $s$ on CIFAR-10 data. Solid lines are the results of numerical experiments and dashed lines are power laws with exponent $\beta$ interpolated in the last decade.