# OpenReview forum: "Locality defeats the curse of dimensionality in convolutional teacher-student scenarios"
_NeurIPS.cc/2021/Conference — NeurIPS 2021 Poster_

### Official Review · Reviewer_qAqm · 2021-07-15

**Rating:** 7
**Confidence:** 4

**Summary:**

In this paper, the authors study the generalization error of kernel regression in a teacher-student scenario where the kernels are obtained as NTK of either one-hidden layer locally-connected NNs or convolutional NNs. They consider the teacher as a Gaussian random field of covariance the same kernel as the student kernel but with different parameters, and proceed to characterize the learning curve exponents. Based on a decomposition of the two kernels in orthonormal basis functions, they use recent (non-rigorous) statistical physics predictions to deduce an exponent that can be shown rigorously to be an upper bound in some cases. This exponent does not depend on the ambient dimension d, only on the patch size, and is the same for the two kernels. The authors deduce that locality is the important property that breaks the curse of dimensionality and notice that adding the invariance by translation only lower the prefactor.
Finally, using a Gaussian universality assumption and a recent framework, they prove a lower bound on the learning curve exponent.  They further check their predictions in numerical experiments on synthetic data.

**Limitations And Societal Impact:**

Yes.

**Main Review:**

Network architecture is believed to play a major role in the good performance of deep neural networks. Yet, theoretical results are severely lacking. In this paper, the authors consider the performance of simple one layer architectures in the kernel regime and show that it beats the curse of dimensionality. I think this is a first concrete step towards understanding the statistical advantages of convolutional neural networks. The paper is well written and the results are discussed. However, I have a few concerns and comments:

(1) While non-trivial, the fact that locality is the important mechanism breaking the curse of dimensionality is not surprising, especially in kernel methods (the learning curve exponent will depend on the eigenvalue decay, which will be set by the local kernels). Given this, it is a bit disappointing that the results are stated for a simplified architecture (non-overlapping patches which makes the RKHS particularly simple, why not use the architecture in the appendix?) and are using non-rigorous arguments. In particular, for these simplified architectures, I believe some classical bounds on KRR will work straightforwardly.

(2) I believe the predictions of Canatar, Bordelon, Pehlevan (2021) matches pretty well numerical simulations in their papers. Is it the case with convolutional kernels in this setting? Also, it would be interesting to test these kernels on real data: for example, with increasing number of samples, showing how the convergence is much faster (while having larger asymptotic error because of smaller RKHS).

(3) Could you explain more why when the student and the teacher are the same, the statistical physics prediction is an upper bound on $\beta$?

Overall, I think the paper is interesting, but (while I understand the appeal of statistical physics proofs) the work would be much more convincing with rigorous bounds.


--- After-rebuttal ---

Thank you for the authors for their detailed response. I think this is a very interesting and timely paper, and for that reason I am raising my rating to 7.

**Time Spent Reviewing:**

3

---

> ### Author Response · Authors · 2021-08-10
> **Authors' reply**
>
> We thank the reviewer for appreciating the relevance of our work and considering it a first concrete step towards understanding the statistical advantage of convolutional neural networks. We reply to the comments in order below.
>
> **(1)** "*It is a bit disappointing that the results are stated for a simplified architecture and are using non-rigorous arguments” (non-overlapping patches which makes the RKHS particularly simple, why not use the architecture in the appendix?).*"
>
> **Reply.** It is key to realise that our theoretical results *do* hold for overlapping-patches architectures as well as nonoverlapping-patches ones. Also, all the experimental results shown refer to kernels with overlapping patches. We will include a sentence at the beginning of Section 4 in the revised manuscript, in order to stress this point further, together with a mention in the `our contributions’ section.
>
> **(2)** "*I believe the predictions of Canatar, Bordelon, Pehlevan matches pretty well numerical simulations. Is it the case in this setting? Also, it would be interesting to test these kernels on real data.*"
>
> **Reply.** We checked the predictions of Canatar et al. in our setting numerically up to training sets of size $10^3$. In this range, their theoretical predictions are in perfect agreement with the learning curves we reported in the paper. For larger training-set sizes, numerical estimations become computationally prohibitive because they require computing and diagonalising kernel matrices with more than $10^{10}$ elements in order to have a sufficient number of modal errors.
>
> Regarding the test on real data: as mentioned in the introduction of our paper, earlier empirical studies (e.g. Ref.[9]) have shown that most of the edge of convolutional architectures with respect to fully-connected ones stems from locality. In addition, we have performed some experiments on MNIST and CIFAR-10. For both datasets, we have considered binary classification tasks (e.g. cats vs dogs or 0 vs 8) converted into regression problems by defining a target function with value $+1$ on one class and $-1$ on the other class.  In both cases, we find as expected that performance is optimal for intermediate values of the filter size $s$. For CIFAR-10, the apparent learning curve exponent is indeed larger at the optimal value of the filter $s*$  (with $\beta(s=s*)\approx 0.1$  whereas for a fully connected,$\beta\approx 0.07$). For MNIST, although performance is clearly better at $s=s*$, the exponent $\beta\approx 0.4$ is hard to distinguish in the two cases. This presumably comes from the fact that MNIST is a simple data set, with a relatively small intrinsic dimension, so it is not subject to the curse of dimensionality --- see discussion in [4]. We will show these data in Appendix in the revised manuscript.
>
>
> **(3)** "*Could you explain more why when the student and the teacher are the same, the statistical physics prediction is an upper bound on beta?*"
>
> **Reply.** When the teacher coincides with the student, the classic bound of Ref.[20] applies, so that $\epsilon(P) \geq B(P)$ (see line 114, apologies for the wrong sign of the inequality, this will be fixed in the revised manuscript). This inequality results from the fact that the mean-squared target coefficients ($\mathbb{E}[c_\rho^2]$ in the manuscript) coincide with the student eigenvalues ($\lambda_\rho$) when student and teacher are equal. Therefore the inequality simply states that the smallest possible error is realised by using the $P$ observations to fit the first $P$ coefficients of the target expansion in the kernel eigenbasis. Inserting the power-law decay for eigenvalues and target coefficients results in the statement that the generalisation error cannot decay faster than some inverse power of $P$, hence the upper bound on $\beta$.
>
> **(4)** "*Overall, I think the paper is interesting, but (while I understand the appeal of statistical physics proofs) the work would be much more convincing with rigorous bounds.*"
>
> **Reply.** It is important to point out that our work considers two realistic assumptions for image-like data (overlapping patches and noiseless labels), as will be clarified in the text.  We are not aware of currently available theoretical tools to control the learning curve exponent rigorously in this setting, i.e. ridgeless regression of Gaussian random functions (the reviewer possibly has in mind source-capacity approaches, but besides the problem of overlapping patches he mentions, in our understanding they do not hold in the absence of measurement noise). We hope that our work will trigger further theoretical studies aimed at pushing these boundaries in similar setups, and we very much agree that it will be valuable. Such studies might benefit from the results on the eigendecomposition of convolutional and local kernels and on the scaling of the tail sum of the eigenvalues provided in our work.
> In addition, let us stress that we do provide a rigorous estimate of the generalisation error in Section 6 for a ridge vanishing as a power law of the size of the training set, under a reasonable Gaussian universality assumption, and that our experiments provide strong validation of the non-rigorous estimate in the ridgless case.

---

### Official Review · Reviewer_vvFT · 2021-07-16

**Rating:** 7
**Confidence:** 3

**Summary:**

This paper studies the two elements of CNNS, 1. local patch; 2. translation-invariant. Under a teacher-student framework for kernel regression, the authors conclude that (under reasonable assumptions), it is the locality of CNNs that defeats the curse of dimensionality. The exponent $\beta$ in the learning curve (with respect to the sample size) is studied theoretically and empirically verified.

**Limitations And Societal Impact:**

Yes

**Main Review:**

** Pros **
1. This paper is very well-written and well-organized. Even though the content addressed is dense, it is still easy to read.
2. The problem studied by the authors is very interesting. The conclusion, i.e., in a teacher-student setting it is the filter locality instead of translation-invariance of convolution that defeats COD, is also significant.
3. The study provides a clearer picture to explain CNN (and in particular, what element of CNN) if the target function can be expressed as a sum of constituent functions depending on locally on the data.

** Cons **
1. Some of the notation in section 2 needs explanation: line 106, what is $c_p$.
2. Equation (5), what is the distribution of the target function $f^*$
3. In line 96, the authors mentioned that "an approximation of $\epsilon$" is derived. Does that mean that (8) and subsequently all the results in the paper (under more general settings) are approximately correct? If that is the case, up to what order?
4. It is interesting to see that weight sharing does not effect the exponent $\beta$. Is there any clue that how weight sharing is going to affect the asymptotic coefficient?

**Review after author response**
The authors have clearly addressed my concerns. My rating remains.

**Time Spent Reviewing:**

5

---

> ### Author Response · Authors · 2021-08-10
> **Authors' reply**
>
> We thank the reviewer for the appreciation shown and for the comments. In particular, we have implemented the suggestions in Cons. 1. and 2. in the revised manuscript. We answer to the other two points in the following (original comments in italic).
>
> **Con 3.** "*In line 96, the authors mentioned that `an approximation of epsilon is derived’. Does that mean that (8) and subsequently all the results in the paper (under more general settings) are approximately correct? If that is the case, up to what order?*"
>
> **Reply.** Eq.(8) was previously calculated with the replica method, which is a non-rigorous technique (rather than an approximation as we erroneously referred to it). The results of replica calculations are often exact and sometimes can be proven rigorously, but that of ridgeless regression is not the case (see reply to point (i) of reviewer 2 (qJrt)). This is why we pay particular attention to supporting our claims with experimental verification and also why we have included Section 6, which refers to regression with a $P$-dependent ridge for which rigorous results are available. This point will be made clearer in the revised manuscript which will include an explanation of the replica method (see also replies to remark 5 of reviewer 5wan and to reviewer qJrt).
>
> **Con 4.** "*It is interesting to see that weight sharing does not affect the exponent beta. Is there any clue that how weight sharing is going to affect the asymptotic coefficient?*"
>
> **Reply.** A recent work (Ref.[26]) has studied rigorously the advantage of weight sharing and translational invariance on the sample complexity. They report that, asymptotically for large training-sets sizes, the gain in sample complexity can be as large as the size of the translation group.  In our framework, this property can be appreciated by comparing each of the 'local’ learning curves with the corresponding 'convolutional’ learning curve, and noticing that the two curves can be made to overlap by scaling the argument $P$ of the convolutional curve with the number of patches. This reduction of complexity due to weight sharing can also be derived from the formalism represented by Equations (8) and (9) (see also reply to point (2) of the 4th reviewer qAqm), as we will indicate in the next version of the manuscript.

---

### Official Review · Reviewer_qJrg · 2021-07-27

**Rating:** 8
**Confidence:** 5

**Summary:**

The authors proved that the generalization error for a CNN can be made independent of the data dimension under certain assumptions. They study the problem in a teacher-student framework and the test error proportional to $P^{-\beta}$ with P to be the training set and $\beta$ to be only dependent on the local kernel size of student network. The authors empirically validate this finding. Furthermore they showed that similar learning curve exponent achieved in the ridgeless case can be achieved in kernel ridge regression with decreasing ridge coefficient.

**Limitations And Societal Impact:**

The authors mentioned there is no negative societal impact which I agree with.

**Main Review:**

The main contribution in Theorem 4.1 is super interesting and solid contribution for this community. I went through most parts of the proofs and it checks out. My main concern is readability of this paper in the current form. With a better readability for the ML audience, I can see this work as a excellent contribution. Please see my detailed comments below.

(a) Change the notation $d^dx$ to $\Pi dx$ for better consistency with literature.
(b) In 1.2, missing reference of "Building Bayesian Neural Networks with Blocks: On Structure, Interpretability and Uncertainty", and some early work of Kondor et al..
(c) The replica method in line 101 should be briefly explained for better readability.
(d) Eq. (8)-(9) are completely without context and makes the reader go check [17, 18]. It needs to be better explained.
(e) what is $c_\rho$ in line 106?
(f) In line 112, how to construct a lattice and sample?
(g) Cite NTK in line 119.
(h)  The sketch of proof for theorem 4.1 is appreciated but it is not clear to be why translation invariant kernel is needed? Insight?
(i) I am not sure the necessity of section 6? Motivation?

Overall it is a excellent contribution but needs rewriting for better readability.

**Time Spent Reviewing:**

8

---

> ### Author Response · Authors · 2021-08-10
> **Authors' reply**
>
> We thank the reviewer for his constructive comments on our work. We appreciate the suggestions to improve readability and we have implemented them in a revised manuscript. See in particular the reply to remark 5 of the first reviewer (5wan) for a sketch of how we plan to address the explanation of Eq.(9) and the replica method. We would also like to comment on points (e) and (f) of the reviewer’s list and we do so in the following (original comments in italic).
>
> **Point (e).** "*The sketch of proof for theorem 4.1 is appreciated but it is not clear to me why translation invariant kernel is needed? Insight?*"
>
> **Reply.** Having a translation-invariant constituent kernel allows us to use the Fourier basis for diagonalisation. We use the specific structure of the Fourier basis, in particular the fact that higher-dimensional plane waves are the product of lower-dimensional plane waves, in two circumstances: when the student has a filter size $s$ which is larger than that of the student $t$ and when considering the overlapping-patches case. Specifically, we use the structure of the Fourier basis in order to express the covariance of the target function as a combination of the student’s eigenfunction.
>
> **Point (i).** "*I am not sure the necessity of Section 6? Motivation?*"
>
> **Reply.** As we stress throughout the manuscript, Equations (8) and (9), which we use to derive the estimation of the error in terms of the tail sum of eigenvalues $B(P)$ (Eq.(10)), are derived with the replica method, which is a non-rigorous technique. Results derived with the replica method have been proven rigorously by other means in several circumstances, but not in this case. In particular, an expression similar to Eq.(8) was proven rigorously in Ref.[21] when the regularisation parameter lambda is larger than the inverse square root of the number of training points $P$, but not in the ridgeless limit. For this reason, we support the results stemming from Eq.(10) with experiments, then show in Section 6 that similar results hold for kernel ridge regression with $P$-dependent ridge (when the rigorous estimate of Ref.[21] applies).

---

### Official Review · Reviewer_5wan · 2021-08-03

**Rating:** 5
**Confidence:** 3

**Summary:**

# Post Rebuttal:

I partly accept the claims of the authors in the rebuttal and have decided to raise my score a little bit to reflect that. I still believe the paper would benefit greatly from clarifications. In particular, I advise the authors to incorporate the changes mentioned in their response to Remarks 1-2 in the rebuttal, as these strengthen the paper in my opinion. As for Remarks 3-4, I disagree with the authors' claim; my point that convolutional neural networks do not provide any benefit over other simpler hypothesis classes in the regime of lazy training still stands and I strongly encourage the authors to omit any reference to neural networks as motivation for studying the problem. I find this reference to be too artificial and somewhat forced.

Lastly, I would like to wish the authors the best of luck in the future.



# Original Review:

The paper studies the expected generalization error when learning in a teacher-student setting where the teacher is a Gaussian random field and the student is a kernel exhibiting a certain convolutional structure. It is shown that under strong assumptions the expected generalization error decays at a rate which does not depend on the input dimension but rather on the filter size of the student. Additionally, experiments are performed to corroborate the theoretical findings in the paper.

**Ethical Concerns:**

I don't foresee any ethical issues that might arise from this paper.

**Limitations And Societal Impact:**

Yes.

**Main Review:**

While the results in the paper provide an interesting example of a learning problem that does not suffer from the curse of dimensionality under certain assumptions, I find the assumptions required to be very specific and highly-stylized. Specifically the results hold for either a data distribution that is uniform on a torus or Gaussian, and the required structure of the teacher further limits the setting. Additionally, the results might only hold when taking the sample size and the regularization parameter in their limit. One motivation for studying the problem as presented by the authors is the connection to neural networks trained in the lazy regime. I find this connection to be weak as neural networks trained in the lazy regime are known to essentially train a linear predictor in high-dimensional space where the features are random and depend on the initialization of the weights of the network (e.g. [1]). In light of this, the results in the paper seem to me to stem from an effective dimensionality reduction provided by the teacher's structure. I don't think this explains why convolutional neural networks are more fit to learn this problem than other types of hypothesis classes and therefore does not address the phenomenon discussed in the beginning of the introduction of image classification on ImageNet.



Moreover, I think the paper would benefit from further clarification as detailed below:

- In line 86 it is written that "the measure p(d^dx)" despite the measure not being previously defined, and no discussion on the nature of this choice is made.
- Eq. (9) lacks explanation. What do the expressions in Eq. (9) represent and why are those relevant?



Typos:

Line 87: rigde -> ridge



Citations used:

[1] Yehudai and Shamir, "On the Power and Limitations of Random Features for Understanding Neural Networks".

**Time Spent Reviewing:**

5

---

> ### Author Response · Authors · 2021-08-10
> **Authors' reply**
>
> We thank the reviewer for his comments and we would like to answer to some of the remarks (original remarks are in italic).
>
> **Remark 1.** "*I find the assumptions required to be very specific and highly-stylized. Specifically the results hold for either a data distribution that is uniform on a torus or Gaussian, and the required structure of the teacher further limits the setting.*"
>
> **Reply.**  The assumption that the data distribution is uniform on a torus is required for our calculation, but it is technical in nature. What is important in the context of regression on Gaussian fields is that the distribution of data is such that the distance between nearest neighbors decreases as $P^{-1/d}$, see e.g. [4] or the literature on kriging. This property will be true for a much broader class of distributions. In the manuscript, we supported this view by performing experiments where data are distributed differently (uniformly on the hypercube and the hypersphere). As expected, they closely match our predictions. We will emphasize this point further in the revised manuscript.
>
> Regarding the choice of the teacher, we disagree that it is a limitation of our setting, we see it instead as a strength. Most theoretical works on neural networks consider very simple models of data (e.g. mixture of Gaussian, or labels that depend linearly on the input), and little has been done to incorporate locality (see our section “related works” for important works in that direction), which is believed to be a key aspect of image-like data. Our set-up has flexibility (by tuning the teacher filter size, referred to as $t$ in our manuscript, we can control the degree of locality of the task) while allowing for asymptotic calculations of training curves which accurately match observations, therefore it allows us to engage quantitatively with the issue of the curse of dimensionality.
>
> **Remark 2.** "*Additionally, the results might only hold when taking the sample size and the regularization parameter in their limit.*"
>
> **Reply.** The approaches we take (based on Ref.[17,21]) work already at finite training set size $P$, as shown in these references, and at finite ridge (which is in fact a simpler case to treat than the ridgeless one). Here, we decided to focus on the asymptotic behavior of these frameworks at large $P$ because it does capture the range of dataset sizes typical for image recognition problem (ImageNet has several hundreds images per class, while CIFAR-10 and MNIST have a few thousands images per class), and the limit of zero regularisation because it is often used in practical applications with images. Those are also the relevant limits to discuss the curse of dimensionality.  Yet, it is straightforward for us to use these formalisms more generally, in combination with our results, to compute finite $P$ effects. In response to the reviewer, we have checked that the formalism of [17] works quantitatively in our setup (see also the answer to the last referee).
>
> **Remark 3.** "*One motivation for studying the problem as presented by the authors is the connection to neural networks trained in the lazy regime. I find this connection to be weak as neural networks trained in the lazy regime are known to essentially train a linear predictor in high-dimensional space where the features are random and depend on the initialization of the weights of the network (e.g. [1]).*"
>
> **Reply.** We disagree with this statement, as it was proven in several papers (Ref.[12-16] in our manuscript) that training an *infinite*-width neural network with proper initialisation of the weights is equivalent to performing kernel regression with the Neural Tangent Kernel (NTK), a limit in which the precise weights chosen at initialisation play no role (as long as they are chosen from a given distribution). This fact is stressed at the end of the introduction of our manuscript (“such students include one-hidden-layer neural networks of diverging width operating in the lazy training regime”) and recalled throughout the third section. Having a diverging width, i.e. an overparametrised network, is crucial in understanding this limit as it is how the random features which depend on the initialisation of the weights lead to a well-defined kernel. In order to avoid possible misunderstanding, we have changed the second sentence of Section 3 (lines 119-129) from “our framework encompasses regression with simple neural networks” to “our framework encompasses regression with simple *overparametrised* neural networks” and also changed “neural networks of diverging width” on line 43 with “overparametrised neural networks” for consistency.
>
> **Remark 4.** "*In light of this, the results in the paper seem to me to stem from an effective dimensionality reduction provided by the teacher's structure. I don't think this explains why convolutional neural networks are more fit to learn this problem than other types of hypothesis classes.*"
>
> **Reply.** We respectfully but strongly disagree with this statement. If our results were a mere consequence of an effective dimension reduction induced by our choice of teacher, then the student would not need a prior on locality to learn that task. But as we show, if the structure of the teacher is crucial to beat the curse of dimensionality, so is the structure of the student. Panels C and D of Figure 1 show precisely that only convolutional/local students having a small filter size are able to leverage the structure of the teacher and beat the curse of dimensionality. In order to strengthen the connection between kernels and neural networks architectures, we will include new figures in the revised manuscript, showing the learning curves for regression with the empirical NTK of overparametrised neural networks. This procedure corresponds exactly to the one mentioned by the reviewer (“train a linear predictor in high-dimensional space where the features are random and depend on the initialization of the weights of the network”). Such new figures display the same behaviour of panels C and D of Figure 1. This is because, although the features associated with the lazy training regime are random, they inherit the structure of the architecture (e.g. locality and/or shift invariance), thus they can be used to explain the edge of convolutional/local architectures over other types of hypothesis classes when learning local functions.
>
> **Remark 5**. "*Moreover, I think the paper would benefit from further clarification as detailed below*"
>
> **Reply.**  Let us thank the reviewer for the suggestions. We have changed “the measure $p(d^dx)$” to “a certain measure $p(d^dx)$”, so as to stress that there is still no specific choice of the data distribution at this stage. Regarding Equation (9), we understand the need for an explanation (see reply to reviewer 2 / qJrg, point (d)), which we will add to the manuscript. Mathematically, the object called $t(P)/P$ is related to the Stieltjes transform of the eigenvalue density of the gram matrix. In practice, it has the role of a threshold for learning (a variant of it was called `signal capture threshold’ in Ref.[21]), in the following sense. If the target function is expanded as a combination of the eigenvectors of the student, the regression problem can be formulated as learning the coefficients of such expansion. These coefficients are reproduced (i.e. learnt) on average if the corresponding eigenvalue is much smaller than the threshold $t(P)$ and are not reproduced otherwise. The expansion of the target function leads to a similar expansion of the generalisation error as a combination of modal contributions. Here the threshold separates modes corresponding to a large eigenvalue, which give a vanishing contribution to the error, from modes corresponding to a small eigenvalue, which give a contribution proportional to the mean-squared coefficient of the target function. The object called $\gamma(P)$, which is a function of $t(P)$, is actually not important in our framework as it always appears in a combination which, in the ridgeless case, tends to $1$ for large dataset size $P$.
>
> Given the amount of explanation required, we propose the inclusion of a new appendix where to move Equations (8) and (9) and their detailed explanation, together with a stylised explanation of the replica method suggested by reviewer 2 / qJrg.

---

### Decision · Program_Chairs · 2021-09-27

**Decision:**

Accept (Poster)

**Comment:**

The paper addresses a topic which has drawn a fair amount of attention recently, namely: if and how do locality and translation-invariance aid convnets in easing the curse of dimensionality. The paper does so by studying a teacher-student framework for kernel regression (see [4]). The reviewers found the implications of the analysis (i.e., 'locality is important') very interesting, and generally appreciated the analytic technique employed in the paper (modulo standard caveats of the replica methods). That said, the paper provoked considerable discussion among the reviewers regarding the acceptable extent to which conclusions drawn based on the kernel ('lazy') regime should be regarded as valid for ReLU networks, and raised several concerns in terms of presentation, clarity. All in all, I think this is a reasonable paper to accept if there is room.